# Gradient Descent can Learn Less Over-parameterized Two-layer Neural Networks on Classification Problems

## Abstract

Recently, several studies have proven the global convergence and generalization abilities of the gradient descent method for two-layer ReLU networks. Most studies especially focused on the regression problems with the squared loss function, except for a few, and the importance of the positivity of the *neural tangent kernel* has been pointed out. However, the performance of gradient descent on classification problems using the logistic loss function has not been well studied, and further investigation of this problem structure is possible. In this work, we demonstrate that the separability assumption using a *neural tangent* model is more reasonable than the positivity condition of the neural tangent kernel and provide a refined convergence analysis of the gradient descent for two-layer networks with smooth activations. A remarkable point of our result is that our convergence and generalization bounds have much better dependence on the network width in comparison to related studies. Consequently, our theory provides a generalization guarantee for less over-parameterized two-layer networks, while most studies require much higher over-parameterization.

## 1 Introduction

In recent years, many studies have been devoted to explaining the great success of over-parameterized neural networks, where the number of parameters is much larger than that needed to fit a given training dataset. This study also treats over-parameterized two-layer neural networks using smooth activation functions and analyzes the convergence and generalization abilities of the gradient descent method for optimizing this type of network.

For over-parameterized two-layer neural networks, Du et al. (2019); Arora et al. (2019); Chizat & Bach (2018a); Mei et al. (2018) showed the global convergence of the gradient descent. These studies are mainly divided into two groups depending on the scaling factor of the output of the networks to which the global convergence property has been demonstrated using different types of proofs. For the scaling factor $1/m$, ($m$: the number of hidden units), Chizat & Bach (2018a); Mei et al. (2018) showed the convergence to the global minimum over probability measures when $m \to \infty$ by utilizing the Wasserstein gradient flow perspective (Nitanda & Suzuki, 2017) on the gradient descent. For the scaling factor $1/m^\beta$ ($\beta < 1$), Du et al. (2019) essentially demonstrated that the kernel smoothing of functional gradients by the *neural tangent kernel* (Jacot et al., 2018; Chizat & Bach, 2018b) has comparable performance with the functional gradient as $m \to \infty$ by making a positivity assumption on the Gram-matrix of this kernel, resulting in the global convergence property. In addition, Arora et al. (2019) provided a generalization bound via a fine-grained analysis of the gradient descent. These studies provide the first steps to understand the role of over-parameterization of neural networks and the gradient descent on regression problems using the squared loss function. For the classification problems with logistic loss, a few studies (Allen-Zhu et al., 2018a; Cao & Gu, 2019a;b) investigated the convergence and generalization abilities of gradient descent under a separability assumption with a suitable model instead of the positivity of the neural tangent kernel. In this study, we further develop this line of research on binary classification problems.

**Our contributions.** We provide fine-grained global convergence and generalization analyses of the gradient descent for two-layer neural networks with smooth activations under a separability

Table 1: Summary of hyperparameter settings and assumptions to achieve an expected $\epsilon$-classification error by gradient descent for binary classifications. The "Separability" column denotes the types of models where a separability assumption is made. $m$ is the number of hidden units, $n$ is the size of the training data, and $T$ is the number of iterations of gradient descent. The notations $\tilde{\Omega}$ and $\tilde{\Theta}$ hide the logarithmic terms in the big-$\Omega$ and -$\Theta$ notations. Smooth activations include *sigmoid, tanh, swish* activations, and several smooth approximations of ReLU. As for Allen-Zhu et al. (2018a); Cao & Gu (2019a;b), we pick up results specialized to two-layer networks.

| | Activation | Separability | Deep | $m$ | $n$ | $T$ |
|---|---|---|---|---|---|---|
| Allen-Zhu et al. (2018a) | ReLU | Smooth Target | yes | $\tilde{\Omega}(\epsilon^{-10})$ | $\Omega(\epsilon^{-4})$ | $\tilde{\Theta}(\epsilon^{-2})$ |
| Cao & Gu (2019a) | ReLU | ReLU NN | yes | $\tilde{\Omega}(\epsilon^{-14})$ | $\tilde{\Omega}(\epsilon^{-4})$ | $\tilde{\Theta}(\epsilon^{-2})$ |
| Cao & Gu (2019b) | ReLU | ReLU NN | yes | $\tilde{\Omega}(\epsilon^{-14})$ | $\tilde{\Omega}(\epsilon^{-2})$ | $\tilde{\Theta}(\epsilon^{-2})$ |
| **This work** | Smooth | Neural Tangent | no | $\Omega(\epsilon^{-1})$ $\tilde{\Theta}(\epsilon^{-3/2})$ | $\tilde{\Omega}(\epsilon^{-4})$ $\tilde{\Omega}(\epsilon^{-2})$ | $\Theta(\epsilon^{-2})$ $\tilde{\Theta}(\epsilon^{-1})$ |

assumption with a sufficient margin using a *neural tangent model*, which is a non-linear model with feature extraction through a neural tangent. We demonstrate that a separability assumption is more suitable than the positivity condition of the neural tangent kernel because (i) the positive neural tangent kernel leads to weak separability and conversely, (ii) separability leads to the positivity of the neural tangent kernel only on a cone spanned by labels, which is very restrictive compared to the whole space. Therefore, the separability condition is rather weak in this sense but it is enough to ensure global convergence for the classification problems. Thus, a significantly better convergence and generalization analyses with respect to network width can be obtained because the positivity of the neural tangent kernel is not required. Consequently, our theory provides a generalization guarantee for less over-parameterized two-layer networks trained by gradient descent, while most existing results relying on the positive neural tangent kernel essentially require quite high over-parameterization. To the best of our knowledge, there are no successful studies for our problem setting (i.e., less over-parameterized two-layer neural networks with smooth activation functions for the classification problems with logistic loss) in the literature. Most studies have focused on highly over-parameterized neural networks with ReLU activation, and less over-parameterized settings have been considered difficult for showing the global convergence property of gradient descent even in the few studies using a separability condition (Allen-Zhu et al., 2018a; Cao & Gu, 2019a;b). However, we note that these studies provided global convergence and generalization analyses of the (stochastic) gradient descent for challenging settings (i.e., deep ReLU networks) by making a similar but different assumption than ours. Thus, our and these studies do not include each other because of the difference of the network structure (i.e., network depth and activation type) and assumptions.

We here describe the main result informally. A neural tangent model is an infinite-dimensional non-linear model using transformed features $(\partial_\theta \sigma(\theta^{(0)\top} x))_{\theta^{(0)} \sim \mu_0}$, where $\sigma$ is a smooth activation and $\mu_0$ is a distribution used to initialize the parameters of the input layer in two-layer neural networks. Theorem 1 states that gradient descent can find an $\epsilon$-accurate solution in terms of the expected classification error for a wide class of over-parameterized two-layer neural networks under a separability assumption using a neural tangent model.

**Theorem 1** (Informal). *Suppose that a given data distribution is separable by a neural tangent model with a sufficient margin under $L_\infty$-constraint. If for any $\epsilon > 0$, the hyperparameters satisfy one of the following*

*(i) $\beta \in [0, 1)$, $m = \Omega(\epsilon^{\frac{-1}{1-\beta}})$, $T = \Theta(\epsilon^{-2})$, $\eta = \Theta(m^{2\beta-1})$, $n = \tilde{\Omega}(\epsilon^{-4})$,*

*(ii) $\beta = 0$, $m = \tilde{\Theta}(\epsilon^{-3/2})$, $T = \tilde{\Theta}(\epsilon^{-1})$, $\eta = \Theta(m^{-1})$, $n = \tilde{\Omega}(\epsilon^{-2})$.*

*then with high probability over the random initialization and choice of samples of size $n$, the gradient descent with a learning rate $\eta$ achieves an expected $\epsilon$-classification error within $T$-iterations.*

**Related work.** A few recent studies (Allen-Zhu et al., 2018a; Cao & Gu, 2019a;b) are closely related to our work because they also treated the logistic loss function. As stated above, problem settings in our and these studies are somewhat different, but we compare our result with those

specialized to two-layer network to show a better property of our problem setting and analyses. Separability assumptions were made on an infinite-width two-layer ReLU network in Cao & Gu (2019a;b) and on a smooth target function in Allen-Zhu et al. (2018a). For generalization analyses, our result exhibits much better dependency on the network width owing to a better problem setting with a fine-grained analysis. Table 1 provides a comparison of the hyperparameter settings of networks and gradient descent in related studies to achieve an expected $\epsilon$-classification error. As evident in Table 1, for more comprehensive sizes of two-layer networks with respect to the network width, our theory ensures the same generalization ability as those of Allen-Zhu et al. (2018a); Cao & Gu (2019a;b). In fact, the network width $\Omega(\epsilon^{-1})$ and $\Omega(\epsilon^{-3/2})$ are sufficient in our setting.

For the stochastic gradient descent for two-layer networks, Brutzkus et al. (2018); Li & Liang (2018) provided generalization analyses. Brutzkus et al. (2018) assumed that datasets are linear separable and this restrictive assumption was relaxed to mixtures of well separated data distributions in Li & Liang (2018). However, the analysis in Li & Liang (2018) is also tailored to only highly over-parameterized settings. Concretely, a very large width $m = \tilde{\Omega}(\epsilon^{-24})$ and the number of samples (iterations) $n = \Theta(T) = \tilde{O}(\epsilon^{-12})$ are required to achieve an expected $\epsilon$-classification error in Li & Liang (2018). In addition, it should be noted that global convergence analyses (Allen-Zhu et al., 2018b; Zou et al., 2018) in terms of optimization without the specification of network size will yield loose generalization bounds because the complexities of neural networks cannot be specified.

Apart from the abovementioned studies, there are many other studies (Brutzkus & Globerson, 2017; Zhong et al., 2017; Tian, 2017; Soltanolkotabi, 2017; Du et al., 2019; Zhang et al., 2018; Arora et al., 2019; Oymak & Soltanolkotabi, 2019; Zhang et al., 2019; Wu et al., 2019) that focus on regression problems but our study focuses on classification problems and demonstrates a better property of gradient descent for over-parameterized networks by utilizing the problem structure of a binary classification. Especially, we show that a separability assumption is more preferable than the positivity condition of the neural tangent kernel (Du et al., 2019; Arora et al., 2019; Zhang et al., 2019; Wu et al., 2019) and show that the required network width can be significantly reduced. Concretely, the required network widths are at least $\Omega(n^6)$ (Du et al., 2019; Wu et al., 2019), $\Omega(n^7\epsilon^{-2})$ (Arora et al., 2019), and $\Omega(n^4)$ (Zhang et al., 2019). Thus, these widths are very large, as compared to our results because sample complexities are generally slower than or equal to $n = \Omega(\epsilon^{-2})$. In addition, proof techniques are also different for the squared loss and the logistic loss functions because the latter function lacks the strong convexity. Thus, we cannot utilize the linear convergence property for the logistic loss and parameters will diverge, which also causes the difficulty of showing better generalization ability without a fine-grained analysis.

## 2 PRELIMINARY

Here, we describe the problem setting for the binary logistic regression and discuss the functional gradients to provide a clear theoretical view of the gradient methods for two-layer neural networks.

### 2.1 PROBLEM SETTING

Let $\mathcal{X} = \mathbb{R}^d$ and $\mathcal{Y}$ be a feature space and the set of binary labels $\{-1, 1\}$, respectively. We denote by $\nu$ a true probability measure on $\mathcal{X} \times \mathcal{Y}$ and by $\nu_n$ an empirical probability measure, deduced from observations $(x_i, y_i)_{i=1}^n$ independently drawn from $\nu$, i.e., $d\nu_n(X, Y) = \sum_{i=1}^n \delta_{(x_i, y_i)}(X, Y) dX dY/n$, where $\delta$ is the Dirac delta function. The marginal distributions of $\nu$ and $\nu_n$ on $X$ are denoted by $\nu^X$ and $\nu_n^X$, respectively. For $\zeta \in \mathbb{R}$ and $y \in \mathcal{Y}$, let $l(\zeta, y)$ be the logistic loss: $\log(1 + \exp(-y\zeta))$. Then, the objective function to be minimized is formalized as follows:

$$\mathcal{L}(\Theta) \stackrel{def}{=} \mathbb{E}_{(X,Y)\sim\nu_n}[l(f_\Theta(X), Y)] = \frac{1}{n}\sum_{i=1}^n l(f_\Theta(x_i), y_i),$$

where $f_\Theta : \mathcal{X} \to \mathbb{R}$ is a two-layer neural network equipped with parameters $\Theta = (\theta_r)_{r=1}^m$. When we consider a function $f_\Theta$ as a variable of the objective function, we denote $\mathcal{L}(f_\Theta) \stackrel{def}{=} \mathcal{L}(\Theta)$.

The two-layer neural network treated in this study is formalized as follows. For parameters $\Theta = (\theta_r)_{r=1}^m$ ($\theta_r \in \mathbb{R}^d$) and fixed constants $(a_r)_{r=1}^m \in \{-1, 1\}^m$:

$$f_\Theta(x) = \frac{1}{m^\beta} \sum_{r=1}^m a_r \sigma(\theta_r^\top x),$$

(1)

where $m$ is the number of hidden units, $\beta$ is an order of the scaling factor, and $\sigma : \mathbb{R} \to \mathbb{R}$ is a smooth activation function such as sigmoid, tanh, swish (Ramachandran et al.)), and other smooth approximations of ReLU. In the training procedure, the parameters $\Theta = (\theta_r)_{r=1}^m$ of the input layer are optimized. This setting is the same as those in Du et al. (2019); Arora et al. (2019); Zhang et al. (2019); Wu et al. (2019), except for the types of activation functions, scaling factor, and loss function.

## 2.2 Functional Gradient

We denote by $L_2(\nu_n^X)$ the function space from $\mathcal{X}$ to $\mathbb{R}$, equipped with the inner product $\langle \cdot, \cdot \rangle_{L_2(\nu_n^X)}$:

$$\langle \phi, \psi \rangle_{L_2(\nu_n^X)} \stackrel{def}{=} \mathbb{E}_{X \sim \nu_n^X} [\phi(X)\psi(X)], \quad \forall \phi, \forall \psi \in L_2(\nu_n^X).$$

Following the tradition in the literature of boosting and kernel methods, we call $L_2(\nu_n^X)$ the function space, although this space is actually an $n$-dimensional space because the cardinality of the support of $\nu_n^X$ is $n$. The key notion to explain the behavior of the gradient descent is the functional gradient in this function space $L_2(\nu_n^X)$. We define the functional gradient at a predictor $f : \mathcal{X} \to \mathbb{R}$ as,

$$\nabla_f \mathcal{L}(f)(x) \stackrel{def}{=} \begin{cases} \partial_\zeta l(\zeta, y_i)|_{\zeta = f(x_i)} & (x = x_i), \\ 0 & \text{(otherwise)}. \end{cases}$$

This is simply a Fréchet differential (functional gradient) in $L_2(\nu_n^X)$. That is, it follows that

$$\mathcal{L}(f + \phi) = \mathcal{L}(f) + \langle \nabla_f \mathcal{L}(f), \phi \rangle_{L_2(\nu_n^X)} + o(\|\phi\|_{L_2(\nu_n^X)}), \quad \forall f, \forall \phi \in L_2(\nu_n^X).$$

Therefore, the functional gradient descent using $\nabla_f \mathcal{L}(f)$ directly optimizes $\mathcal{L}$ in a function space $L_2(\nu_n^X)$ and converges to a global minimum because the objective function $\mathcal{L}$ is convex with respect to a function $f$. However, because $\nabla_f \mathcal{L}(f)$ contains no information regarding the unseen data, this method is meaningless in terms of generalization. Thus, some smoothing techniques are required to guarantee the generalization. The gradient descent method for two-layer neural networks can be recognized as a type of kernel-smoothed functional gradient using the *neural tangent kernel* (Jacot et al., 2018), and this perspective is significantly useful in showing the global convergence because it characterizes the behavior of the vanilla gradient descent in a function space.

## 3 Brief Review of Functional Gradient Methods

Functional gradient methods have been mainly studied for gradient boosting (Mason et al., 1999; Friedman, 2001) and kernel methods (Kivinen et al., 2004; Smale & Yao, 2006; Ying & Zhou, 2006; Raskutti et al., 2014; Wei et al., 2017) in the machine learning community, but more recently, it has been found to be useful in explaining the behavior of gradient descent for neural networks (Jacot et al., 2018; Chizat & Bach, 2018b; Du et al., 2019; Allen-Zhu et al., 2018b; Arora et al., 2019). Our analysis is also heavily based on the functional gradient perspective of gradient descent. Thus, we briefly review the functional gradient methods.

In gradient boosting, $\nabla_f \mathcal{L}(f)$ is approximated by finding a similar function in weak learners $\mathcal{G}$:

$$\phi_f \in \underset{\phi \in \mathcal{G}}{\arg\max} \langle \nabla_f \mathcal{L}(f), \phi \rangle_{L_2(\nu_n^X)}$$

(2)

and the gradient method in a function space is performed using a descent direction $-\phi_f$. This approximation is a type of smoothing of functional gradients. In kernel methods, this smoothing procedure is realized by using the *kernel smoothing* technique:

$$T_k \nabla_f \mathcal{L}(f) \stackrel{def}{=} \mathbb{E}_{\nu_n^X}[\nabla_f \mathcal{L}(f)(X)k(X, \cdot)] = \frac{1}{n} \sum_{i=1}^n \nabla_f \mathcal{L}(f)(x_i)k(x_i, \cdot),$$

(3)

where $k$ is a kernel function. It should be noted that this kernel smoothing (3) is a special case of gradient boosting (2) because of the following equation:

$$\frac{T_k \nabla_f \mathcal{L}(f)}{\|T_k \nabla_f \mathcal{L}(f)\|_{\mathcal{H}_k}} \in \underset{\|\phi\|_{\mathcal{H}_k} \leq 1}{\operatorname{argmax}} \langle \nabla_f \mathcal{L}(f), \phi \rangle_{L_2(\nu_n^X)},$$

where $(\mathcal{H}_k, \langle, \rangle_{\mathcal{H}_k})$ is the reproducing kernel Hilbert space associated with a kernel $k$. When this kernel smoothing well approximates a functional gradient $\nabla_f \mathcal{L}(f)$ and satisfies

$$\langle \nabla_f \mathcal{L}(f), T_k \nabla_f \mathcal{L}(f) \rangle_{L_2(\nu_n^X)} \geq \exists \mu \|\nabla_f \mathcal{L}(f)\|_{L_2(\nu_n^X)}^2, \tag{4}$$

the kernel-smoothed functional gradient descent $f^+ \leftarrow f - \eta T_k \nabla_f \mathcal{L}(f)$ performs like the pure functional gradient descent, leading to the global convergence property because it tends to a stationary point in a function space, which is simply a global minimum.

Recently, several studies (Jacot et al., 2018; Chizat & Bach, 2018b; Du et al., 2019; Allen-Zhu et al., 2018b; Arora et al., 2019) implicitly or explicitly pointed out that the gradient descent for neural networks is essentially recognized as an approximation to the kernel-smoothed functional gradient method using a *neural tangent kernel* (NTK) (Jacot et al., 2018):

$$k_{NTK}(x, x') \overset{def}{=} \mathbb{E}_{\theta^{(0)} \sim \mu_0}[\partial_\theta \sigma(\theta^{(0)\top} x)^\top \partial_\theta \sigma(\theta^{(0)\top} x')], \tag{5}$$

where $\mu_0$ is a distribution to initialize the parameters of the input layer in this setting. In most proofs using NTK, the global convergence property has been demonstrated by showing the condition (4) from the positivity of the Gram-matrix $H^\infty \overset{def}{=} (k_{NTK}(x_i, x_j))_{i,j=1}^n$ and the similarity between the gradient descent and the kernel-smoothed functional gradient with NTK when $m \to \infty$. This is a reason why very high over-parameterization is generally required in related studies.

In this study, we found that the positivity of the Gram-matrix of NTK is not required on binary classification problems and a separability assumption, which is a weaker condition than the positivity, is enough for global convergence. Consequently, we can give global convergence and generalization guarantees to a gradient method for less over-parameterized two-layer neural networks.

## 4 GLOBAL CONVERGENCE ANALYSIS OF THE GRADIENT METHOD

The following is an update rule of gradient descent with respect to the input parameters $\Theta = (\theta_r)_{r=1}^m$:

$$\Theta^{(t+1)} \leftarrow \Theta^{(t)} - \eta \nabla_\Theta \mathcal{L}(\Theta^{(t)}), \tag{6}$$

where $\nabla_\Theta \mathcal{L}(\Theta^{(t)}) = (\partial_{\theta_r} \mathcal{L}(\Theta^{(t)}))_{r=1}^m$ and $\eta > 0$ is a learning rate. We here make the assumption:

**Assumption 1.**

**(A1)** *Assume that* $\operatorname{supp}(\nu^X) \subset \{x \in \mathcal{X} \mid \|x\|_2 \leq 1\}$. *Let* $\sigma$ *be a* $\mathcal{C}^2$-class *function and there exist* $K_1, K_2 > 0$ *s.t.* $\|\sigma'\|_\infty \leq K_1$ *and* $\|\sigma''\|_\infty \leq K_2$.

**(A2)** *A distribution* $\mu_0$ *on* $\mathbb{R}^d$ *used for the initialization of* $\theta_r$ *has a sub-Gaussian tail bound:* $\exists A, \exists b > 0$ *such that* $\mathbb{P}_{\theta^{(0)} \sim \mu_0}[\|\theta^{(0)}\|_2 \geq t] \leq A \exp(-bt^2)$.

**(A3)** *Assume that the number of hidden units* $m \in \mathbb{Z}_+$ *is an even number. Constant parameters* $(a_r)_{r=1}^m$ *and parameters* $\Theta^{(0)} = (\theta_r^{(0)})_{r=1}^m$ *are initialized symmetrically:* $a_r = 1$ *for* $r \in \{1, \ldots, \frac{m}{2}\}$, $a_r = -1$ *for* $r \in \{\frac{m}{2} + 1, \ldots, m\}$, *and* $\theta_r^{(0)} = \theta_{r+\frac{m}{2}}^{(0)}$ *for* $r \in \{1, \ldots, \frac{m}{2}\}$, *where the initial parameters* $(\theta_r^{(0)})_{r=1}^{\frac{m}{2}}$ *are independently drawn from a distribution* $\mu_0$.

**(A4)** *Assume that there exist* $\rho > 0$ *and a measurable function* $v : \mathbb{R}^d \to \{w \in \mathbb{R}^d \mid \|w\|_2 \leq 1\}$ *such that the following inequality holds: for* $\forall (x, y) \in \operatorname{supp}(\nu) \subset \mathcal{X} \times \mathcal{Y}$,

$$y \left\langle \partial_\theta \sigma(\theta^{(0)\top} x), v(\theta^{(0)}) \right\rangle_{L_2(\mu_0)} = y \mathbb{E}_{\theta^{(0)} \sim \mu_0}[\partial_\theta \sigma(\theta^{(0)\top} x)^\top v(\theta^{(0)})] \geq \rho. \tag{7}$$

**Remark.** Clearly, many activation functions (sigmoid, tanh, and smooth approximations of ReLU such as swish) satisfy the assumption **(A1)**. Typical distributions, including the Gaussian distribution, satisfy **(A2)**. The purpose of the symmetrized initialization **(A3)** is to bound the initial value of the loss function $\mathcal{L}(\Theta^{(0)})$ uniformly over the number of hidden units $m$. This initialization leads to $f_{\Theta^{(0)}}(x) = 0$, resulting in $\mathcal{L}(\Theta^{(0)}) = \log(2)$. Assumption **(A4)** implies the separability of a dataset using the *neural tangent* model. We next discuss the validity of this assumption.

## 4.1 Separability Assumption (A4) by the Neural Tangent

The explicit feature representation: $x \to \partial_\theta \sigma(\theta^{(0)\top} x)$, of NTK (5) is called the *neural tangent*, which is a non-linear feature extraction from $\mathcal{X}$ to an infinite-dimensional space. That is, assumption **(A4)** ensures the separability of the transformed data $(\partial_\theta \sigma(\theta_r^{(0)\top} x), y)$ through the neural tangent for $(x, y) \in \text{supp}(\nu)$ with a margin $\rho$ in an infinite-dimensional space by the weight: $v(\theta^{(0)}) d\mu_0$. We remark that this assumption is somewhat weaker than the positivity assumption on the Gram-matrix of NTK and is satisfied in many cases by the universal approximation ability of the neural tangent models. In addition, we remark that the separability of the training dataset instead of $\text{supp}(\nu)$ is enough to guarantee global convergence only for empirical risk minimization.

**Theoretical comparison of kernel assumptions.** In previous studies (Du et al., 2019; Arora et al., 2019; Zhang et al., 2019; Wu et al., 2019) the positivity of the Gram-matrix $H^\infty = (k_{NTK}(x_i, x_j))_{i,j=1}^n$ was required to ensure the condition (4). Here, we remark that the assumption **(A4)** is weaker than this positivity condition in the following sense.

**Proposition 1.** *(i) Assume $H^\infty \succeq \lambda_0 I_n$ and $\|\sigma'\|_\infty \leq K_1$, then there exists a measurable map $v : \mathbb{R}^d \to \{w \in \mathbb{R}^d \mid \|w\|_2 \leq 1\}$ such that $\forall i \in \{1, \ldots, n\}$,*

$$y_i \left\langle \partial_\theta \sigma(\theta^{(0)\top} x_i), v(\theta^{(0)}) \right\rangle_{L_2(\mu_0)} \geq \frac{\lambda_0}{nK_1}.$$

*(ii) Suppose assumption **(A4)** holds, then $\sum_{i,j=1}^n \xi_i H^\infty \xi_j \geq \rho^2 \|\xi\|_2^2$, $(\forall \xi \in \{(\alpha_i y_i)_{i=1}^n \mid \alpha_i \geq 0\})$.*

As seen in Proposition 1-(i), the positivity $H^\infty \succeq \lambda_0 I_n$ leads to weak separability with a margin of $O(\lambda_0/n)$ on the training dataset. Conversely, from Proposition 1-(ii), the separability with a margin of $\rho$ leads to the positivity $\rho^2$ of $H^\infty$ only on a cone spanned by the labels: $\{(\alpha_i y_i)_{i=1}^n \mid \alpha_i \geq 0\}$. Because this cone is very restrictive, this limited positivity is much weaker than the positivity on the whole space. However, we found that this limited positivity is sufficient to ensure the global convergence of the gradient descent for binary classification problems with logistic loss. Indeed, from Proposition 3, the positivity of $H^\infty$ is required only along the functional gradients: $\nabla_f \mathcal{L}(f_\Theta)(x_i) = \partial_\zeta l(f_\Theta(x_i), y_i)$, and these functional gradients are always contained in this limited space, unlike the squared loss function. This is a reason why the positivity of NTK is not required for the binary classification problem with logistic loss. Thus, a much better convergence and generalization ability can be shown for logistic loss than the previous results that relied on the positivity of $H^\infty$ because the positivity of NTK on the whole space is redundant and a separability condition provides a better positivity only on a required small space. Concretely, from Proposition 1-(i), we can immediately check a deteriorated convergence result depending on the positivity of $H^\infty$ by replacing $\rho$ in Theorem 2 with $O(\lambda_0/n)$, producing $m = O(n\lambda_0^{-1}\epsilon^{-1})$ when $\beta = 0$.

**Remark.** For regression problem, Allen-Zhu et al. (2018b;c); Zou et al. (2018); Oymak & Soltanolkotabi (2019); Zou & Gu (2019) make a different separation where examples are away from each other: $\|x_i - x_j\|_2 \geq \rho$. Zou & Gu (2019) shows that this assumption is essentially same as the positivity of NTK. Thus, it also completely differs from (A4) as shown in Proposition 1

**Universal approximation property of neural tangent models.** We consider the case where all feature vectors have a common bias term: $x = (x^0, \ldots, x^{d-1}, s) \in \mathcal{X}$ ($s > 0$ is a sufficiently small constant for a bias term). In this case, we can easily confirm that neural tangent models include typical two-layer infinite-width neural networks with activation $\sigma'$: $\mathbb{E}[w(\theta^{(0)})\sigma'(\theta^{(0)\top} x)]$ by setting $v(\theta^{(0)}) = (0, \ldots, 0, w(\theta^{(0)}))$, where $w$ is a real-valued function. Thus, Assumption (A.4) with a certainly positive constant $\rho$ is satisfied as long as a data distribution is separable by an infinite-width two-layer network with mild weights $w(\theta)$. Moreover, we note that these networks have the universal

approximation property (Hornik, 1991; Sonoda & Murata, 2017), so that there are a lot of examples such that the assumption **(A4)** is satisfied.

## 4.2 MAIN RESULTS

We here define the $L_1$-norm of the functional gradient, which measures the convergence.

$$\|\nabla_f \mathcal{L}(f_\Theta)\|_{L_1(\nu_n^X)} \overset{def}{=} \frac{1}{n} \sum_{i=1}^{n} |\partial_\zeta l(f_\Theta(x_i), y_i)| = \frac{1}{2n} \sum_{i=1}^{n} |y_i - 2p_\Theta(Y = 1 \mid x_i) + 1|.$$

Here, $\zeta$ is the first variable of $l$ and $p_\Theta(Y = 1 \mid x) = \frac{1}{1+\exp(-f_\Theta(x))}$ is a conditional probability on $Y = 1$, defined by $f_\Theta$. Because $\|\nabla_f \mathcal{L}(f_\Theta)\|_{L_1(\nu_n^X)}$ is simply a gap between the labels and conditional label probabilities of the model, this norm is a reasonable measure for the binary classification problems. The following is the first main result to ensure global convergence.

**Theorem 2** (Global Convergence). *Suppose Assumption 1 holds. Set $K = K_1^4 + 2K_1^2 K_2 + K_1^4 K_2^2$. For $\forall \beta \in [0, 1)$, $\forall \delta \in (0, 1)$ and $\forall m \in \mathbb{Z}_+$ such that $m \geq \frac{16K_1^2}{\rho^2} \log \frac{2n}{\delta}$, consider gradient descent (6) with a learning rate of $0 < \eta \leq \min \left\{ m^\beta, \frac{4m^{2\beta-1}}{K_1^2+K_2} \right\}$ and the number of iterations $T \leq \left\lfloor \frac{m\rho^2}{32\eta K_2^2 \log(2)} \right\rfloor$. Then, it follows that with probability at least $1-\delta$ over the random initialization,*

$$\frac{1}{T} \sum_{t=0}^{T-1} \|\nabla_f \mathcal{L}(f_{\Theta^{(t)}})\|_{L_1(\nu_n^X)}^2 \leq \frac{16\log(2)}{\rho^2 T} \left( \frac{m^{2\beta-1}}{\eta} + K \right). \tag{8}$$

We here derive a corollary, which states that an arbitrary small empirical classification error can be achievable by appropriately setting $\eta, m$, and $T$ as follows. From Markov's inequality,

$$\mathbb{P}_{(X,Y) \sim \nu_n}[Y f_{\Theta^{(t)}}(X) \leq 0] \leq 2\|\nabla_f \mathcal{L}(f_\Theta)\|_{L_1(\nu_n^X)},$$

where we used the following relationship:

$$0.5 |y_i - 2p_\Theta(Y = 1|x_i) + 1| \geq (1 + \exp(\gamma))^{-1} \iff y_i f_\Theta(x_i) \leq \gamma.$$

Thus, a convergence rate of $\|\nabla_f \mathcal{L}(f_{\Theta^{(t)}})\|_{L_1(\nu_n^X)}^2$ leads to a rate of empirical classification error.

**Corollary 1.** *Suppose the same assumptions as in Theorem 2 hold. If for $\forall \epsilon, \delta > 0$, the hyperparameters satisfy*

$$\beta \in [0, 1), \ m = \Omega(\rho^{\frac{-1}{1-\beta}} \epsilon^{\frac{-1}{1-\beta}}), \ T = \Omega(\rho^{-2}\epsilon^{-2}), \ \eta = \Theta(m^{2\beta-1}),$$

*then with probability at least $1 - \delta$, gradient descent (6) with a learning rate of $\eta$ finds a parameter $\Theta^{(t)}$ satisfying $\mathbb{P}_{(X,Y) \sim \nu_n}[Y f_{\Theta^{(t)}}(X) \leq 0] \leq \epsilon$ within $T$-iterations.*

The Landau notations are applied with respect to $\epsilon, \rho \to 0$. Utilizing this theorem, we can show the convergence of the loss function which leads to a better result at the price of slight increase of $m$.

**Theorem 3.** *Suppose the same assumptions in Theorem 2 hold. Then there exists a uniform constant $C > 0$ such that for $0 < \forall \alpha \leq \frac{\rho}{4K_2}$, with probability at least $1 - \delta$,*

$$\frac{1}{T} \sum_{t=0}^{T-1} \mathcal{L}(\Theta^{(t)}) \leq C \left( \frac{1}{T} + \frac{\alpha^2 m}{\eta T} + \exp\left( -\frac{\alpha \rho m^{1-\beta}}{4} \right) + \frac{\alpha^2}{\rho} \sqrt{\frac{m}{\eta T}} + \frac{1}{\rho} \sqrt{\frac{\eta T}{m}} \right).$$

**Corollary 2.** *Suppose the same assumptions as in Theorem 3 hold. Then there exists a uniform constant $C > 0$ and the following statement holds. If for any $\epsilon > 0$, the hyperparameters satisfy*

$$\beta = 0, \ m = \Theta(\rho^{-2}\epsilon^{-3/2} \log(1/\epsilon)), \ T_* = \Theta(\rho^{-2}\epsilon^{-1} \log^2(1/\epsilon)), \ \eta = \Theta(m^{-1}),$$

*then with probability $1 - \delta$, $\frac{1}{T} \sum_{t=0}^{T-1} \mathcal{L}(\Theta^{(t)}) \leq C \left( \epsilon + \frac{1}{\rho^2 T} \log^2 \left( \frac{1}{\epsilon} \right) \right)$ for $0 < \forall T \leq T_*$.*

This corollary is obtained immediately from Theorem 3 by setting $\alpha = \Theta(\rho \epsilon^{3/2})$. The convergence of classification error also derived from the corollary because $\mathcal{L}(f_\theta) \geq \|\nabla_f \mathcal{L}(f_{\Theta^{(t)}})\|_{L_1(\nu_n^X)}$ and it also derives a sharper bound on the distance $\|\Theta^{(T)} - \Theta^{(0)}\|_2$.

**Proposition 2.** *Suppose the same assumption and consider the same hyperparameter setting as in Corollary 2. Then, there exists a uniform constant $C$ such that with probability $1 - \delta$,*

$$\|\Theta^{(T)} - \Theta^{(0)}\|_2 \le C\epsilon^{3/4} \log^2(\rho^{-2}\epsilon^{-1}).$$

Moreover, by combining Theorem 2 and Corollary 2 with the well-known margin bound (Koltchinskii & Panchenko, 2002; Mohri et al., 2012; Shalev-Shwartz & Ben-David, 2014) on the expected classification error and by specifying the Rademacher complexity of the function class attained by gradient descent, we can obtain the second main result for the generalization ability of gradient descent.

**Theorem 4** (Generalization Bound). *Suppose Assumption 1 holds. Set $K = K_1^4 + 2K_1^2 K_2 + K_1^4 K_2^2$. Fix $\forall \gamma > 0$. Consider the gradient descent (6) with a general hyperparameter setting in Theorem 2 or a specific setting in Corollary 2, with $\delta \in (0,1)$. For these cases, we set parameters $C_{\eta,m,T}$ and $D_{\eta,m,T}$ as follows:*

*(Former case)* $\quad C_{\eta,m,T} = \rho^{-1} T^{-1/2} \left( m^{\beta - \frac{1}{2}} \eta^{-1/2} + \sqrt{K} \right), \quad D_{\eta,m,T} = \sqrt{\eta T}$

*(Latter case)* $\quad C_{\eta,m,T} = \epsilon + \rho^{-2} T^{-1} \log^2(1/\epsilon), \quad D_{\eta,m,T} = \epsilon^{3/4} \log^2(\rho^{-2}\epsilon^{-1}).$

*Then, there exists a uniform constant $C > 0$ and it follows that with probability at least $1 - 3\delta$ over a random initialization and random choice of dataset $S$,*

$$\min_{t \in \{0,\ldots,T-1\}} \mathbb{P}_{(X,Y)\sim\nu}[Y f_{\Theta^{(t)}}(X) \le 0] \le C(1 + \exp(\gamma))C_{\eta,m,T} + 3\sqrt{\frac{\log(2/\delta)}{2n}}$$
$$+ C\gamma^{-1} m^{\frac{1}{2}-\beta} D_{\eta,m,T}(1 + K_1 + K_2)\sqrt{\frac{d}{n}\log\left(n(1 + K_1 + K_2)(\log(m/\delta) + D_{\eta,m,T}^2)\right)}. \quad (9)$$

*Moreover, when $\sigma$ is convex and $\sigma(0) = 0$, we can avoid the dependence with respect to the dimension $d$. With probability at least $1 - 3\delta$ over a random initialization and random choice of dataset $S$*

$$\min_{t \in \{0,\ldots,T-1\}} \mathbb{P}_{(X,Y)\sim\nu}[Y f_{\Theta^{(t)}}(X) \le 0] \le C(1 + \exp(\gamma))C_{\eta,m,T} + 3\sqrt{\frac{\log(2/\delta)}{2n}}$$
$$+ \frac{CK_1 m^{\frac{1}{2}-\beta}}{\gamma\sqrt{n}}\left(D_{\eta,m,T} + \sqrt{\frac{\log(Am/\delta)}{b}}\right). \quad (10)$$

This theorem provides an upper-bound on an expected classification error with high probability for a network obtained by gradient descent within $T$-iterations. There is a trade-off between the optimization and complexity terms in (9), (10) with respect to $\eta, m,$ and $T$. However, there are several choices of these hyperparameters to achieve a desired precision $\epsilon$ of the expected classification error.

**Corollary 3.** *Suppose the same assumptions as in Theorem 4 hold. If for any $\epsilon > 0$, the hyperparameters satisfy one of the following*

*(i)* $\beta \in [0,1)$, $m = \Omega(\rho^{\frac{-2}{1-\beta}}\epsilon^{\frac{-1}{1-\beta}})$, $T = \Omega(\rho^{-2}\epsilon^{-2})$, $\eta = \Theta(\rho^{-2}\epsilon^{-2}T^{-1}m^{2\beta-1})$, $n = \tilde{\Omega}(\rho^{-2}\epsilon^{-4})$,

*(ii)* $\beta = 0$, $m = \Theta(\rho^{-2}\epsilon^{-3/2}\log(1/\epsilon))$, $T = \Theta(\rho^{-2}\epsilon^{-1}\log^2(1/\epsilon))$, $\eta = \Theta(m^{-1})$, $n = \tilde{\Omega}(\epsilon^{-2})$,

*then with probability at least $1 - \delta$, the gradient descent (6) with a learning rate of $\eta$ finds a parameter $\Theta^{(t)}$ satisfying $\mathbb{P}_{(X,Y)\sim\nu}[Y f_{\Theta^{(t)}}(X) \le 0] \le \epsilon$ within $T$-iterations.*

This corollary can be immediately proven by substituting the concrete values of $\beta, m, T, \eta,$ and $n$ into the right hand side of inequalities (9), (10) and by checking that this hyperparameter setting satisfies the conditions required in Theorem 4.

From Corollary 3, for an arbitrary small $\epsilon > 0$, an expected $\epsilon$-classification error is achieved by the gradient descent within $O(1/\epsilon^2)$-or $O(1/\epsilon)$-iterations when the transformed data distribution by the neural tangent: $(\partial_\theta(\theta_r^{(0)\top}\cdot))_{\theta\sim\mu_0}$ is separable in the infinite-dimensional space $L_2(\mu_0)$ under the $L_\infty$-constraint with a sufficient margin $\rho$. In comparison to the result in Allen-Zhu et al. (2018a); Cao & Gu (2019a;b), which also derived a generalization bound by making a similar separability assumption using a ReLU network or a smooth target function instead of the tangent model, our

result has much better dependency on the network width and can explain the generalization ability for a less over-parameterized two-layer network, as summarized in Table 1. However, we note that their theories cover deeper networks and are not included in our theory because of the difference of the problem settings (e.g., network depth and the type of activation functions). To reduce the network width, the best choice of $\beta \in [0, 1)$ is $\beta = 0$ for the first setting, leading to a small network width $m = \Omega(\epsilon^{-1})$. We note that an arbitrary large width is also covered by this result.

### 4.3 PROOF IDEA

In this section, we provide a proof idea for Theorems 2, 3, and 4. We introduce two important propositions that connect the gradient descent with the functional gradient descent to justify an intuitive explanation in Section 3. Proposition 3-(i) states that the gradient descent method is certainly similar to the kernel smoothed gradient methods by the neural tangent kernel when a parameter $\Theta$ is sufficiently close to a stationary point and the learning rate $\eta$ is sufficiently small, and Proposition 3-(ii) states that the loss landscape is almost convex with respect to the parameter when $f_\Theta$ is sufficiently close to a stationary point in the function space. Here, We define an approximated neural tangent kernel: $k_\Theta$ depending on the parameters $\Theta$ as follows:

$$k_\Theta(x, x') \stackrel{def}{=} \partial_\Theta f_\Theta(x)^\top \partial_\Theta f_\Theta(x'). \tag{11}$$

This kernel is actually an approximation to the vanilla NTK as follows:

$$m^{2\beta-1} k_{\Theta^{(0)}}(x, x') \rightarrow k_{NTK}(x, x') \ (m \rightarrow \infty).$$

**Proposition 3.** *Suppose assumption* **(A1)** *holds and* $\beta \in [0, 1)$.
*(i) We set* $\Theta^+ = \Theta - \eta \nabla_\Theta \mathcal{L}(\Theta)$ *and* $K = K_1^2 + 2K_2 + K_1^2 K_2^2$. *If* $\eta \leq m^\beta$, *then*

$$\left| \mathcal{L}(f_{\Theta^+}) - \left( \mathcal{L}(f_\Theta) - \eta \left\langle \nabla_f \mathcal{L}(f_\Theta), T_{k_\Theta} \nabla_f \mathcal{L}(f_\Theta) \right\rangle_{L_2(\nu_n^X)} \right) \right| \leq \frac{\eta^2 K}{2m^{2\beta-1}} \|\nabla_\Theta \mathcal{L}(\Theta)\|_2^2.$$

*(ii) It follows that for* $\Theta = (\theta_r)_{r=1}^m$ *and* $\Theta^* = (\theta_r^*)_{r=1}^m$, $(\theta_r, \theta_r^* \in \mathbb{R}^d)$,

$$\mathcal{L}(\Theta) + \nabla_\Theta \mathcal{L}(\Theta)^\top (\Theta^* - \Theta) \leq \mathcal{L}(\Theta^*) + \frac{K_2}{m^\beta} \|\nabla_f \mathcal{L}(f_\Theta)\|_{L_1(\nu_n^X)} \|\Theta^* - \Theta\|_2^2.$$

The next proposition states that the kernel smoothed gradients have comparable optimization ability to pure functional gradients in terms of minimizing the $L_1$-norm around an initial parameter $\Theta^{(0)}$. We define the $\| \cdot \|_{2,1}$-norm in the parameter space $\Theta = (\theta_r)_{r=1}^m$ as $\|\Theta\|_{2,1} \stackrel{def}{=} \sum_{r=1}^m \|\theta_r\|_2$.

**Proposition 4.** *Suppose Assumption 1 holds. For* $\forall \delta \in (0, 1)$ *and* $\forall m \in \mathbb{Z}_+$, *such that* $m \geq \frac{16K_1^2}{\rho^2} \log \frac{2n}{\delta}$, *the following statement holds with probability at least* $1 - \delta$ *over the random initialization of* $\Theta^{(0)} = (\theta_r^{(0)})_{r=1}^m$. *If* $\|\Theta - \Theta^{(0)}\|_{2,1} \leq \frac{m\rho}{4K_2}$, *then*

$$\left\langle \nabla_f \mathcal{L}(f_\Theta), T_{k_\Theta} \nabla_f \mathcal{L}(f_\Theta) \right\rangle_{L_2(\nu_n^X)} \geq \frac{\rho^2}{16m^{2\beta-1}} \|\nabla_f \mathcal{L}(f_\Theta)\|_{L_1(\nu_n^X)}^2.$$

This proposition is specialized to binary classification problems because the positivity of the Gram-matrix is required for regression problems to make a similar statement as discussed earlier.

Combining these two propositions, we can connect the gradient descent with the functional gradient descent and show the global convergence (Theorem 2) which with the almost convexity (Proposition 3-(ii)) is also used to derive the convergence rate for the loss function (Theorem 3). In addition, applying a well-known result (Koltchinskii & Panchenko, 2002), we can derive a skeleton of a generalization bound (Theorem 4) composed of the margin distribution and Rademacher complexity. As for the margin distribution, the upper bound is obtained by Theorem 2 and 3. As for ways to bound Rademacher complexity, please see the Appendix.

## 5 CONCLUSION

In this paper, we have provided refined global convergence and generalization analyses of the gradient descent for two-layer neural networks with smooth activations on binary classification problems. The

key in our analysis is the separability assumption by a neural tangent model and we have explained the reasonability of this assumption in comparison to the positivity of NTK. Consequently, theoretical justification has been provided for less over-parameterized neural networks. However, our theory is restricted to the *deterministic* gradient descent and two-layer networks; hence, its possible extensions to stochastic gradient descent and deep neural networks are also interesting. Another possible future study is to relax the positivity assumption on the Gram-matrix for regression problems by utilizing our theory and conducting further investigations of the trajectory of gradient descent, such as the shortest pass analysis (Oymak & Soltanolkotabi, 2018).

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

APPENDIX

## A RELATIONSHIP BETWEEN KERNEL ASSUMPTIONS

*Proof of Proposition 1.* We here prove the statement (i). Since $H^\infty$ is invertible, we set $w = (H^\infty)^{-1}(y_1, \cdots, y_n)^\top$ and set

$$v(\theta^{(0)}) = \sum_{j=1}^n \partial_\theta \sigma(\theta^{(0)\top} x_j) w_j.$$

Then, we get

$$y_i \left\langle \partial_\theta \sigma(\theta^{(0)\top} x_i), v(\theta^{(0)}) \right\rangle_{L_2(\mu_0)} = y_i H_{i*}^\infty w = 1.$$

We can bound the norm of $\|v(\theta^{(0)})\|_2$ as follows:

$$\begin{aligned}
\|v(\theta^{(0)})\|_2 &\leq \sum_{j=1}^n \|\partial_\theta \sigma(\theta^{(0)\top} x_j)\|_2 |w_j| \\
&\leq \left\| \left( \|\partial_\theta \sigma(\theta^{(0)\top} x_j)\|_2 \right)_{j=1}^n \right\|_2 \|w\|_2 \\
&\leq \sqrt{n} K_1 \|w\|_2 \\
&\leq \frac{n K_1}{\lambda_0}.
\end{aligned}$$

Thus, by resetting $v(\theta^{(0)}) \leftarrow \frac{\lambda_0 v(\theta^{(0)})}{n K_1}$, we conclude the statement (i).

We next prove the statement (ii). For $\xi = (\alpha_i y_i)_{i=1}^n$ ($\alpha_i > 0$),

$$
\begin{aligned}
\sum_{i,j=1}^n \xi_i H^\infty \xi_j &= \sum_{i,j=1}^n \mathbb{E}_{\theta^{(0)} \sim \mu_0}[\xi_i \partial_\theta(\theta^{(0)\top} x_i)^\top \partial_\theta(\theta^{(0)\top} x_j) \xi_j] \\
&= \mathbb{E}_{\theta^{(0)} \sim \mu_0} \left[ \left\| \sum_{i=1}^n \xi_i \partial_\theta(\theta^{(0)\top} x_i) \right\|_2^2 \right] \\
&\geq \mathbb{E}_{\theta^{(0)} \sim \mu_0} \left[ \left( \sum_{i=1}^n \xi_i \partial_\theta(\theta^{(0)\top} x_i)^\top v(\theta^{(0)}) \right)^2 \right] \\
&\geq \left( \mathbb{E}_{\theta^{(0)} \sim \mu_0} \left[ \sum_{i=1}^n \xi_i \partial_\theta(\theta^{(0)\top} x_i)^\top v(\theta^{(0)}) \right] \right)^2 \\
&= \left( \sum_{i=1}^n \alpha_i \mathbb{E}_{\theta^{(0)} \sim \mu_0} \left[ y_i \partial_\theta(\theta^{(0)\top} x_i)^\top v(\theta^{(0)}) \right] \right)^2 \\
&\geq \rho^2 \left( \sum_{i=1}^n \alpha_i \right)^2 \\
&\geq \rho^2 \sum_{i=1}^n \alpha_i^2 \\
&= \rho^2 \|\xi\|_2^2,
\end{aligned}
$$

where we used $\|v(\theta^{(0)})\|_2 \leq 1$ for the first inequality, the convexity of $\|\cdot\|_2^2$ and Jensen's inequality for the second inequality, Assumption **(A4)** for the third inequality, and $\|\cdot\|_2 \leq \|\cdot\|_1$ for the last inequality. Thus, we finish the proof of the statement (ii). $\qquad\square$

## B   AUXILIARY RESULTS

In this section, we introduce several existing results for proving our statements. We first describe the Hoeffding's inequality.

**Lemma 1** (Hoeffding's inequality). *Let $Z, Z_1, \ldots, Z_m$ be i.i.d. random variables taking values in $[-a, a]$ for $a > 0$. Then, for any $\epsilon > 0$, we get*

$$
\mathbb{P}\left[ \left| \frac{1}{m} \sum_{r=1}^m Z_r - \mathbb{E}[Z] \right| > \epsilon \right] \leq 2 \exp\left( -\frac{\epsilon^2 m}{2a^2} \right).
$$

We here define the *covering number* as follows.

**Definition 1** (Covering Number). *Let $(V, \|\cdot\|)$ a metric space. A subset $\hat{U} \subset V$ is called an $\epsilon$-(proper) cover of $V$ if for $\forall v \in V$, there exists $v' \in \hat{U}$ such that $\|v - v'\| < \epsilon$. Then, $\epsilon$-covering number $\mathcal{N}(V, \epsilon, \|\cdot\|)$ of $V$ is defined as the cardinaly of the smallest $\epsilon$-cover of $V$, that is,*

$$
\mathcal{N}(V, \epsilon, \|\cdot\|) \overset{def}{=} \min\{|\hat{U}| \mid \hat{U} \text{ is an } \epsilon\text{-cover of } V\}.
$$

The following lemma provide a bound on the Rademacher complexity by Dudley's integral. For a real-valued function class $\mathcal{F}$ over $\mathcal{X}$ and a subset $X = (x_i)_{i=1}^n$, $\mathcal{F}|_X$ is defined as $\{(h(x_i))_{i=1}^n \in \mathbb{R}^n \mid h \in \mathcal{F}\} \subset \mathbb{R}^n$, and $\mathcal{F}|_X$ can be equipped with $\|\cdot\|_\infty$-norm over $X$.

**Lemma 2** (Bartlett et al. (2017)). *Let $\mathcal{F}$ be a class of real-valued functions taking values in $[0, 1]$ from $\mathcal{X}$ and assume $0 \in \mathcal{F}$. For examples $\forall X = (x_i)_{i=1}^n$ of size $n$, we get*

$$
\Re(\mathcal{F}|_X) \leq \inf_{\alpha > 0} \left( 4\alpha + \frac{12}{\sqrt{n}} \int_\alpha^1 \sqrt{\log(\mathcal{N}(\mathcal{F}|_X, \epsilon, \|\cdot\|_\infty))} d\epsilon \right).
$$

Note that we reformulate the statement in Lemma 2 from $\|\cdot\|_2$-covering to $\|\cdot\|_\infty$-covering.

## C  PROOFS OF MAIN RESULTS

In this section, we give an outline of proofs of Theorem 2 and 4.

**Global convergence.**    We first introduce two important propositions which connects gradient methods with functional gradient methods. The following proposition states that gradient descent methods become similar to kernel smoothed gradient methods by the neural tangent kernel when a parameter $\Theta$ is sufficiently close to a stationary point and a learning rate $\eta$ is sufficiently small.

**Proposition 5** (Restatement of Proposition 3).  *Suppose assumption* **(A1)** *holds and* $\beta \in [0, 1)$.
*(i) We set* $\Theta^+ = \Theta - \eta \nabla_\Theta \mathcal{L}(\Theta)$ *and* $K = K_1^2 + 2K_2 + K_1^2 K_2^2$. *If* $\eta \le m^\beta$, *then*

$$\left| \mathcal{L}(f_{\Theta^+}) - \left( \mathcal{L}(f_\Theta) - \eta \left\langle \nabla_f \mathcal{L}(f_\Theta), T_{k_\Theta} \nabla_f \mathcal{L}(f_\Theta) \right\rangle_{L_2(\nu_n^X)} \right) \right| \le \frac{\eta^2 K}{2m^{2\beta-1}} \|\nabla_\Theta \mathcal{L}(\Theta)\|_2^2.$$

*(ii) It follows that for* $\Theta = (\theta_r)_{r=1}^m$ *and* $\Theta^* = (\theta_r^*)_{r=1}^m$, $(\theta_r, \theta_r^* \in \mathbb{R}^d)$,

$$\mathcal{L}(\Theta) + \nabla_\Theta \mathcal{L}(\Theta)^\top (\Theta^* - \Theta) \le \mathcal{L}(\Theta^*) + \frac{K_2}{m^\beta} \|\nabla_f \mathcal{L}(f_\Theta)\|_{L_1(\nu_n^X)} \|\Theta^* - \Theta\|_2^2.$$

The next proposition states that kernel smoothed gradients have comparable optimization ability to pure functional gradients in terms of the $L_1$-norm around an initial parameter $\Theta^{(0)}$. We introduce the $\|\cdot\|_{2,1}$-norm in the parameter space $\Theta = (\theta_r)_{r=1}^m$ as $\|\Theta\|_{2,1} \overset{def}{=} \sum_{r=1}^m \|\theta_r\|_2$.

**Proposition 6** (Restatement of Proposition 4).  *Suppose Assumption 1 holds. For* $\forall \delta \in (0, 1)$ *and* $\forall m \in \mathbb{Z}_+$, *such that* $m \ge \frac{16K_1^2}{\rho^2} \log \frac{2n}{\delta}$, *the following statement holds with probability at least* $1 - \delta$ *over the random initialization of* $\Theta^{(0)} = (\theta_r^{(0)})_{r=1}^m$. *If* $\|\Theta - \Theta^{(0)}\|_{2,1} \le \frac{m\rho}{4K_2}$, *then*

$$\left\langle \nabla_f \mathcal{L}(f_\Theta), T_{k_\Theta} \nabla_f \mathcal{L}(f_\Theta) \right\rangle_{L_2(\nu_n^X)} \ge \frac{\rho^2}{16m^{2\beta-1}} \|\nabla_f \mathcal{L}(f_\Theta)\|_{L_1(\nu_n^X)}^2.$$

This proposition is specialized to binary classification problems because the positivity of the Gram-matrix is needed for regression problems in order to make a similar statement as discussed earlier.

We specify the possible number of iterations of gradient descent (6) such that $\Theta^{(t)}$ can remain in the neighborhood: $\{\Theta \mid \|\Theta - \Theta^{(0)}\|_2 \le \frac{\sqrt{m}\rho}{4K_2}\} \subset \{\Theta \mid \|\Theta - \Theta^{(0)}\|_{2,1} \le \frac{m\rho}{4K_2}\}$.

**Proposition 7.**  *Suppose Assumption* **(A1)** *and* **(A3)** *hold. Consider gradient descent (6) with learning rate* $0 < \eta < \frac{4m^{2\beta-1}}{K_1^2 + K_2}$ *and the number of iterations* $T \in \mathbb{Z}_+$. *Then,*

$$\frac{1}{T} \sum_{t=0}^{T-1} \|\nabla_\Theta \mathcal{L}(\Theta^{(t)})\|_2^2 \le \frac{2}{\eta T} \log(2). \tag{12}$$

*Especially, we get* $\|\Theta^{(T)} - \Theta^{(0)}\|_2 \le \sqrt{2\eta T \log(2)}$. *As a result, gradient descent can be performed for* $\left\lfloor \frac{m\rho^2}{32\eta K_2^2 \log(2)} \right\rfloor$-*iterations within* $\{\Theta \mid \|\Theta - \Theta^{(0)}\|_2 \le \frac{\sqrt{m}\rho}{4K_2}\} \subset \{\Theta \mid \|\Theta - \Theta^{(0)}\|_{2,1} \le \frac{m\rho}{4K_2}\}$.

This proposition provides a bound on the distance $\|\Theta^{(T)} - \Theta^{(0)}\|_2$, but we note that this bound will be further sharpened after showing the convergence of the loss function (see Proposition 2). From Proposition 5, 6, and 7, we notice that the gradient descent for $\mathcal{L}(\Theta)$ performs like a pure functional gradient descent up to $O\left(\frac{m\rho^2}{\eta}\right)$-iterations, resulting in significant decrease of loss functions. We next provide the proof of Theorem 2 based on this idea.

*Proof of Theorem 2.*  From Proposition 7, the assumption in Proposition 6 regarding $\Theta$ is satisfied. Thus, Proposition 5 and 6 state that for $t \in \{0, \dots, T-1\}$,

$$\mathcal{L}(f_{\Theta^{(t+1)}}) \le \mathcal{L}(f_{\Theta^{(t)}}) - \frac{\eta\rho^2}{16m^{2\beta-1}} \|\nabla_f \mathcal{L}(f_{\Theta^{(t)}})\|_{L_1(\nu_n^X)}^2 + \frac{\eta^2 K}{2m^{2\beta-1}} \|\nabla_\Theta \mathcal{L}(\Theta^{(t)})\|_2^2.$$

Summing this inequality over $t \in \{0, \ldots, T-1\}$ and multiplying by $\frac{4m^{2\beta-1}}{\eta\rho^2 T}$, we have

$$\frac{1}{T}\sum_{t=0}^{T-1}\|\nabla_f \mathcal{L}(f_{\Theta^{(t)}})\|^2_{L_1(\nu_n^X)} \leq \frac{16m^{2\beta-1}}{\eta\rho^2 T}\mathcal{L}(f_{\Theta^{(0)}}) + \frac{8\eta K}{\rho^2 T}\sum_{t=0}^{T-1}\|\nabla_\Theta \mathcal{L}(\Theta^{(t)})\|^2_2.$$

Applying $\mathcal{L}(\Theta^{(0)}) = \log(2)$ and inequality (12), we complete the proof. $\qquad\square$

*Proof of Theorem 3.* We set $\tau^* = \left(\alpha a_r v(\theta_r^{(0)})\right)_{r=1}^m$ and $\Theta^* = \Theta^{(0)} + \tau^*$. Clearly, we have

$$\|\Theta^* - \Theta^{(0)}\|_2 \leq \alpha\sqrt{m}. \tag{13}$$

As shown in Proposition 5, we get

$$\left| f_{\Theta^*}(x) - \nabla_\Theta f_{\Theta^{(0)}}(x)^\top (\Theta^* - \Theta^{(0)}) \right| \leq \frac{K_2}{m^\beta}\|\tau^*\|_2^2 \leq \alpha^2 K_2 m^{1-\beta}.$$

In addition, as shown in Proposition 6, since $m \geq \frac{16K_1^2}{\rho^2}\log\frac{2n}{\delta}$, the inequality (21) holds with probability at least $1 - \delta$. Hence, we have for $\forall i \in \{1, \ldots, n\}$,

$$y_i f_{\Theta^*}(x_i) \geq y_i \nabla_\Theta f_{\Theta^{(0)}}(x)^\top (\Theta^* - \Theta^{(0)}) - \alpha^2 K_2 m^{1-\beta}$$

$$= \frac{y_i \alpha}{m^\beta}\sum_{r=1}^m \partial_\theta \sigma(\theta_r^{(0)\top} x_i)^\top v(\theta_r^{(0)}) - \alpha^2 K_2 m^{1-\beta}$$

$$\geq \frac{\alpha\rho m^{1-\beta}}{2} - \alpha^2 K_2 m^{1-\beta} \geq \frac{\alpha\rho m^{1-\beta}}{4}.$$

Thus, the loss at a reference point $\Theta^*$ can be bounded as follows:

$$\mathcal{L}(\Theta^*) \leq \frac{1}{n}\sum_{i=1}^n \log\left(1 + \exp(-y_i f_{\Theta^*}(x_i)\right) \leq \exp\left(-y_i f_{\Theta^*}(x_i)\right) \leq \exp\left(-\frac{\alpha\rho m^{1-\beta}}{4}\right). \tag{14}$$

From Theorem 2, Proposition 5-(ii), Proposition 7 and inequalities (13), (14), it follows that $\exists C_1, \exists C_2 > 0, \forall T \leq T$,

$$\frac{1}{T}\sum_{t=0}^{T-1}\left(\mathcal{L}(\Theta^{(t)}) + \nabla_\Theta \mathcal{L}(\Theta^{(t)})^\top(\Theta^* - \Theta^{(t)})\right)$$

$$\leq \mathcal{L}(\Theta^*) + \frac{K_2}{m^\beta T}\sum_{t=0}^{T-1}\|\nabla_f \mathcal{L}(f_{\Theta^{(t)}})\|_{L_1(\nu_n^X)}\|\Theta^* - \Theta^{(t)}\|_2^2$$

$$\leq \mathcal{L}(\Theta^*) + \frac{K_2}{m^\beta T}\sum_{t=0}^{T-1}\|\nabla_f \mathcal{L}(f_{\Theta^{(t)}})\|_{L_1(\nu_n^X)}\max_{t \in \{0,\ldots,T-1\}}\|\Theta^* - \Theta^{(t)}\|_2^2$$

$$\leq \mathcal{L}(\Theta^*) + \frac{2K_2}{m^\beta\sqrt{T}}\sqrt{\sum_{t=0}^{T-1}\|\nabla_f \mathcal{L}(f_{\Theta^{(t)}})\|^2_{L_1(\nu_n^X)}\max_{t \in \{0,\ldots,T-1\}}\left(\|\Theta^* - \Theta^{(0)}\|_2^2 + \|\Theta^{(0)} - \Theta^{(t)}\|_2^2\right)}$$

$$\leq \mathcal{L}(\Theta^*) + \frac{C_2}{\rho\sqrt{\eta T m}}(\alpha^2 m + \eta T)$$

$$\leq \exp\left(-\frac{\alpha\rho m^{1-\beta}}{4}\right) + \frac{C_2}{\rho}\left(\alpha^2\sqrt{\frac{m}{\eta T}} + \sqrt{\frac{\eta T}{m}}\right). \tag{15}$$

Finally, we bound the average of $\nabla_\Theta \mathcal{L}(\Theta^{(t)})^\top (\Theta^{(t)} - \Theta^*)$. Because $-2a^\top b = \|a\|_2^2 + \|b\|_2^2 - \|a+b\|_2^2$ for real vectors $a, b$, we get by setting $a = -\eta \nabla_\Theta \mathcal{L}(\Theta^{(t)})$ and $b = \Theta^{(t)} - \Theta^*$,

$$
\begin{aligned}
\frac{1}{T} \sum_{t=0}^{T-1} \nabla_\Theta \mathcal{L}(\Theta^{(t)})^\top (\Theta^{(t)} - \Theta^*) &= \frac{1}{2\eta T} \sum_{t=0}^{T-1} (\eta^2 \|\nabla_\Theta \mathcal{L}(\Theta^{(t)})\|_2^2 + \|\Theta^{(t)} - \Theta^*\|_2^2 - \|\Theta^{(t+1)} - \Theta^*\|_2^2) \\
&= \frac{\eta}{2T} \sum_{t=0}^{T-1} \|\nabla_\Theta \mathcal{L}(\Theta^{(t)})\|_2^2 + \frac{1}{2\eta T} \|\Theta^{(0)} - \Theta^*\|_2^2 \\
&\leq \frac{\log(2)}{T} + \frac{\alpha^2 m}{2\eta T}.
\end{aligned}
$$

Thus, we get that $\exists C > 0$,

$$
\frac{1}{T} \sum_{t=0}^{T-1} \mathcal{L}(\Theta^{(t)}) \leq C \left( \frac{1}{T} + \frac{\alpha^2 m}{\eta T} + \exp\left(-\frac{\alpha \rho m^{1-\beta}}{4}\right) + \frac{\alpha^2}{\rho} \sqrt{\frac{m}{\eta T}} + \frac{1}{\rho} \sqrt{\frac{\eta T}{m}} \right).
$$

$\square$

We next prove Proposition 2 that gives a sharper bound on $\|\Theta^{(T)} - \Theta^{(0)}\|$.

*Proof of Proposition 2.* Let $L \in \mathbb{Z}_+$ be a positive integer such that for $T = O(\rho^{-2} \epsilon^{-1} \log^2(1/\epsilon))$, $2^L \leq T < 2^{L+1}$. Clearly, $L \leq \log_2 T$. From Corollary 2, we get for $l \in \{1, \ldots, L\}$

$$
\frac{1}{2^{l-1}} \sum_{t=2^{l-1}}^{2^l-1} \mathcal{L}(\Theta^{(t)}) \leq \frac{2}{2^l} \sum_{t=0}^{2^l-1} \mathcal{L}(\Theta^{(t)}) \leq 2C \left( \epsilon + 2^{-l} \rho^{-2} \log^2(1/\epsilon) \right).
$$

Therefore, there exist $2^{l-1} \leq \exists s_l < 2^l$ for $l \in \{1, \ldots, L\}$ such that

$$
\mathcal{L}(\Theta^{(s_l)}) \leq 2C \left( \epsilon + 2^{-l} \rho^{-2} \log^2(1/\epsilon) \right).
$$

From the similar argument to the proof of Proposition 7, we get for $a < b \in \mathbb{Z}_+$, $\sum_{t=a}^{b} \|\nabla_\Theta \mathcal{L}(\Theta^{(t)})\|_2 \leq \sqrt{2(b-a+1)\eta^{-1} \mathcal{L}(\Theta^{(t)})}$.

Thus, it follows that since $s_1 = 1$, $\|\nabla_\Theta \mathcal{L}(\Theta^{(0)})\|_2 \leq \sqrt{m} K_1$ by (19), and $2^{l+1} - 2^{l-1} + 1 \leq 2^{l+1}$,

$$
\begin{aligned}
\sum_{t=0}^{T-1} \|\nabla_\Theta \mathcal{L}(\Theta^{(t)})\|_2 &\leq \|\nabla_\Theta \mathcal{L}(\Theta^{(0)})\|_2 + \sum_{l=1}^{L-1} \sum_{t=s_l}^{s_{l+1}} \|\nabla_\Theta \mathcal{L}(\Theta^{(t)})\|_2 + \sum_{t=s_L}^{T-1} \|\nabla_\Theta \mathcal{L}(\Theta^{(t)})\|_2 \\
&\leq \sqrt{m} K_1 + \sum_{l=1}^{L} \sqrt{2^3 C \eta^{-1}(2^l \epsilon + \rho^{-2} \log^2(1/\epsilon))} \\
&\leq \sqrt{m} K_1 + \log_2(T) \sqrt{2^3 C \eta^{-1}(T\epsilon + \rho^{-2} \log^2(1/\epsilon))}.
\end{aligned}
$$

Hence, by setting specific values of $\eta, T$, and $m$ in Corollary 2, we get that $\exists C' > 0$,

$$
\begin{aligned}
\|\Theta^{(t)} - \Theta^{(0)}\|_2 \leq \eta \sum_{t=0}^{T-1} \|\nabla_\Theta \mathcal{L}(\Theta^{(t)})\|_2 &\leq \eta \sqrt{m} K_1 + \log_2(T) \sqrt{2^3 C \eta(T\epsilon + \rho^{-2} \log^2(1/\epsilon))} \\
&\leq C' \epsilon^{3/4} \log^2(\rho^{-2} \epsilon^{-1}).
\end{aligned}
$$

$\square$

**Generalization bound.** A generalization bound can be derived by utilizing the standard analysis of the Rademacher complexity (Koltchinskii & Panchenko, 2002). We here introduce a function class to be measured by the Rademacher complexity. Let $l_\gamma(v)$ ($\gamma > 0$) be the *ramp* loss:

$$
l_\gamma(v) \stackrel{def}{=} \begin{cases} 1 & (v < 0), \\ 1 - v/\gamma & (0 \leq v \leq \gamma), \\ 0 & (v > \gamma). \end{cases}
$$

Then, a class of all possible ramp losses over $\mathcal{X} \times \mathcal{Y}$ attained by the gradient descent (6) up to $T$-iterations is defined as follows: $\Omega_{\eta,m,T} \overset{def}{=} \left\{ \Theta \mid \|\Theta - \Theta^{(0)}\|_2 \leq D_{\eta,T,m} \right\}$,

$$\mathcal{F}_{\eta,m,T}^{\gamma} \overset{def}{=} \left\{ l_\gamma(y f_\Theta(x)) : \mathcal{X} \times \mathcal{Y} \to [0,1] \mid \Theta \in \Omega_{\eta,m,T} \right\}.$$

Here, $D_{\eta,T,m}$ is set to be $\sqrt{2\eta T \log(2)}$ when considering a general hyperparameter setting in Theorem 2 and is set to be a sharper bound in Proposition 2: $\Theta(\epsilon^{3/4} \log^2(\rho^{-2}\epsilon^{-1}))$ when considering a specific hyperparameter setting in that proposition.

For a given dataset $S = (x_i, y_i)_{i=1}^{n}$, the Rademacher complexity is defined by $\Re(\mathcal{F}_{\eta,m,T}^{\gamma}|_S) \overset{def}{=} n^{-1}\mathbb{E}[\sup_{h \in \mathcal{F}_{\eta,m,T}^{\gamma}} \sum_{i=1}^{n} \epsilon_i h(x_i, y_i)]$, where the expectation is taken over the Rademacher random variables $(\epsilon_i)_{i=1}^{n}$ which are i.i.d. with probabilities $\mathbb{P}[\epsilon_i = 1] = \mathbb{P}[\epsilon_i = -1] = 0.5$. The following well-known result (Koltchinskii & Panchenko, 2002; Mohri et al., 2012; Shalev-Shwartz & Ben-David, 2014) provides a bound on the expected classification error based on the empirical margin distribution and the Rademacher complexity. The empirical margin distribution for $S$ is defined as the ratio of examples satisfying $y_i f_\Theta(x_i) \leq \gamma$ in $S$.

**Lemma 3** (Koltchinskii & Panchenko (2002); Mohri et al. (2012); Shalev-Shwartz & Ben-David (2014)). *Let $\forall n \in \mathbb{Z}_+$, $\forall \gamma > 0$, $\forall \eta > 0$, $\forall m \in \mathbb{Z}_+$, $\forall T \in \mathbb{Z}_+$, and $\forall \delta \in (0,1)$. Then, with probability at least $1 - \delta$ over the random choice of $S$ of size $n$, every $\Theta \in \Omega_{\eta,m,T}$ satisfies*

$$\mathbb{P}_{(X,Y)\sim\nu}[Y f_\Theta(X) \leq 0] \leq \mathbb{P}_{(X,Y)\sim\nu_n}[Y f_\Theta(X) \leq \gamma] + 2\Re(\mathcal{F}_{\eta,m,T}^{\gamma}|_S) + 3\sqrt{(2n)^{-1}\log(2/\delta)}. \tag{16}$$

To instantiate this bound, we have to provide upper bounds on the empirical margin distribution and the Rademacher complexity. We first give a bound on the Rademacher complexity.

**Proposition 8.** *Suppose Assumption (A1) and (A2) hold. Let $\forall \gamma > 0$, $\forall \eta > 0$, $\forall m \in \mathbb{Z}_+$, $\forall T \in \mathbb{Z}_+$, $\forall \delta \in (0,1)$, and $\forall S$ be examples of size $n$. Then, there exists a uniform constant $C > 0$ such that with probability at least $1 - \delta$ with respect to the initialization of $\Theta^{(0)}$,*

$$\Re(\mathcal{F}_{\eta,m,T}^{\gamma}|_S) \leq C\gamma^{-1}m^{\frac{1}{2}-\beta}D_{\eta,m,T}(1+K_1+K_2)\sqrt{\frac{d}{n}\log\left(n(1+K_1+K_2)(\log(m/\delta)+D_{\eta,m,T}^2)\right)}.$$

*Moreover, when $\sigma$ is convex and $\sigma(0) = 0$, we can avoid the dependence with respect to the dimension $d$. With probability at least $1 - \delta$ over a random initialization of $\Theta^{(0)}$,*

$$\Re(\mathcal{F}_{\eta,m,T}^{\gamma}|_S) \leq \frac{8K_1 m^{\frac{1}{2}-\beta}}{\gamma\sqrt{n}}\left(D_{\eta,m,T} + \sqrt{\frac{\log(Am/\delta)}{b}}\right).$$

*Proof of Theorem 4.* We prove this theorem by instantiating inequality (16). Let $(\Theta^{(t)})_{t=0}^{T-1}$ be a sequence obtained by the gradient descent (6). Because $(\Theta^{(t)})_{t=0}^{T-1}$ is contained in $\Omega_{\eta,m,T}$, as indicated in Proposition 7, inequality (16) holds for this sequence. As for the Rademacher complexity in (16), we can utilize Proposition 8. Thus, the resulting problem is to prove the convergence of the empirical margin distribution: $\mathbb{P}_{(X,Y)\sim\nu_n}[Y f_{\Theta^{(t)}}(X) \leq \gamma]$. We here give its upper-bound below.

$$0.5\left|y_i - 2p_\Theta(Y=1|x_i) + 1\right| \geq (1 + \exp(\gamma))^{-1} \iff y_i f_\Theta(x_i) \leq \gamma.$$

Therefore, from Markov's inequality,

$$\mathbb{P}_{(X,Y)\sim\nu_n}[Y f_{\Theta^{(t)}}(X) \leq \gamma] = \mathbb{P}_{(X',Y')\sim\nu_n}\left[\frac{1}{2}|Y' - 2p_\Theta(Y=1|X') + 1| \geq \frac{1}{1+\exp(\gamma)}\right]$$
$$\leq (1 + \exp(\gamma))\|\nabla_f \mathcal{L}(f_\Theta)\|_{L_1(\nu_n^X)}.$$

Combining this inequality with Lemma 3, then for $\forall t \in \{0, \ldots, T-1\}$,

$$\mathbb{P}_{(X,Y)\sim\nu}[Y f_{\Theta^{(t)}}(X) \leq 0] \leq (1 + \exp(\gamma))\|\nabla_f \mathcal{L}(f_{\Theta^{(t)}})\|_{L_1(\nu_n^X)} + 2\Re(\mathcal{F}_{\eta,m,T}^{\gamma}|_S) + 3\sqrt{\frac{\log(2/\delta)}{2n}}.$$

Noting that $\eta$, $m$, and $T$ satisfy the conditions in Theorem 2, we can complete the proof by taking the average over $t \in \{0, \ldots, T-1\}$ and applying Proposition 8 and Theorem 2. $\qquad\square$

# D   PROOFS FOR GLOBAL CONVERGENCE

## D.1   PROOF OF PROPOSITION 5

*Proof of Proposition 5.* We first show the smoothness of $f_\Theta(x)$ with respect to $\Theta$ for $\forall x \in \mathcal{X}$, $(\|x\|_2 \le 1)$. Noting that $\nabla_\Theta^2 f_\Theta(x) = \mathrm{diag}\left(\frac{1}{m^\beta} a_r \sigma''(\theta_r^\top x) xx^\top\right)_{r=1}^m$, we get for $\tau = (\tau_r)_{r=1}^m$ such that $\sum_{r=1}^m \|\tau_r\|_2^2 = 1$ $(\tau_r \in \mathbb{R}^d)$,

$$
\begin{aligned}
\left|\tau^\top \nabla_\Theta^2 f_\Theta(x) \tau\right| &= \left|\sum_{r=1}^m \tau_r^\top \partial_{\theta_r}^2 f_\Theta(x) \tau_r\right| \\
&\le \frac{1}{m^\beta} \sum_{r=1}^m \left|\sigma''(\theta_r^\top x)\right| \left|\tau_r^\top x\right|^2 \\
&\le \frac{K_2}{m^\beta} \sum_{r=1}^m \|\tau_r\|_2^2 \\
&= \frac{K_2}{m^\beta}.
\end{aligned}
$$

This means that for $\tau = (\tau_r)_{r=1}^m$, $(\tau_r \in \mathbb{R}^d)$,

$$
\left|f_{\Theta+\tau}(x) - (f_\Theta(x) + \nabla_\Theta f_\Theta(x)^\top \tau)\right| \le \frac{K_2}{m^\beta} \|\tau\|_2^2 = \frac{K_2}{m^\beta} \sum_{r=1}^m \|\tau_r\|_2^2. \tag{17}
$$

Let us define $g_x(\tau)$ as the second-order term of Taylor's expansion of $f_\Theta(x)$ with respect to $\Theta$:

$$
f_{\Theta+\tau}(x) = f_\Theta(x) + \nabla_\Theta f_\Theta(x)^\top \tau + g_x(\tau).
$$

From the inequality (17), we have $|g_x(\tau)| \le \frac{K_2 \|\tau\|_2^2}{m^\beta}$. Then, by the smoothness of $l(\zeta, y)$ with respect to $\zeta$ and $|\partial_\zeta^2 l(\zeta, y)| \le 1/4$, we get

$$
\begin{aligned}
\Big|l(f_{\Theta+\tau}(x), y) &- (l(f_\Theta(x), y) + \partial_\zeta l(f_\Theta(x), y)(\nabla_\Theta f_\Theta(x)^\top \tau + g_x(\tau)))\Big| \\
&\le \frac{1}{4} \left|\nabla_\Theta f_\Theta(x)^\top \tau + g_x(\tau)\right|^2 \\
&\le \frac{1}{2}\left(\|\nabla_\Theta f_\Theta(x)\|_2^2 + \frac{K_2^2 \|\tau\|_2^2}{m^{2\beta}}\right) \|\tau\|_2^2.
\end{aligned}
$$

By the triangle inequality, we get

$$
\begin{aligned}
\big|l(f_{\Theta+\tau}(x), y) &- \big(l(f_\Theta(x), y) + \partial_\zeta l(f_\Theta(x), y)\nabla_\Theta f_\Theta(x)^\top \tau\big)\big| \\
&\le |\partial_\zeta l(f_\Theta(x), y) g_x(\tau)| + \frac{1}{2}\left(\|\nabla_\Theta f_\Theta(x)\|_2^2 + \frac{K_2^2\|\tau\|_2^2}{m^{2\beta}}\right)\|\tau\|_2^2 \\
&\le \frac{1}{2}\left(\|\nabla_\Theta f_\Theta(x)\|_2^2 + \frac{2K_2}{m^\beta} + \frac{K_2^2\|\tau\|_2^2}{m^{2\beta}}\right)\|\tau\|_2^2 \\
&\le \frac{1}{2}\left(\frac{K_1^2}{m^{2\beta-1}} + \frac{2K_2}{m^\beta} + \frac{K_2^2\|\tau\|_2^2}{m^{2\beta}}\right)\|\tau\|_2^2, \tag{18}
\end{aligned}
$$

where for the second inequality, we used $|\partial_\zeta l(\zeta, y)| \le 1$ and for the last inequality, we used

$$
\|\nabla_\Theta f_\Theta(x)\|_2^2 = \sum_{r=1}^m \left\|\frac{1}{m^\beta}\sigma'(\theta_r^\top x)x\right\|_2^2 \le \sum_{r=1}^m \frac{1}{m^{2\beta}}|\sigma'(\theta_r^\top x)|_2^2 \le \frac{K_1^2}{m^{2\beta-1}}.
$$

We here set $\tau = -\eta \nabla_\Theta \mathcal{L}(\Theta)$. The right hand side of (18) is upper bounded by

$$
\frac{1}{2}\left(\frac{K_1^2}{m^{2\beta-1}} + \frac{2K_2}{m^\beta} + \frac{\eta^2 K_1^2 K_2^2}{m^{4\beta-1}}\right)\|\tau\|_2^2.
$$

because

$$
\begin{aligned}
\|\nabla_\Theta \mathcal{L}(\Theta)\|_2^2 &= \sum_{r=1}^m \|\partial_{\theta_r}\mathcal{L}(\Theta)\|_2^2 \\
&= \sum_{r=1}^m \left\|\frac{1}{n}\sum_{i=1}^n \partial_\zeta l(f_\Theta(x_i),y_i)\partial_{\theta_r}f_\Theta(x_i)\right\|_2^2 \\
&\leq \sum_{r=1}^m \left(\frac{1}{n}\sum_{i=1}^n |\partial_\zeta l(f_\Theta(x_i),y_i)|\|\partial_{\theta_r}f_\Theta(x_i)\|_2\right)^2 \\
&\leq \left(\frac{1}{n}\sum_{i=1}^n |\partial_\zeta l(f_\Theta(x_i),y_i)|\right)^2 \sum_{r=1}^m \left(\max_{j\in\{1,\dots,n\}}\|\partial_{\theta_r}f_\Theta(x_j)\|_2\right)^2 \\
&= \|\nabla_f \mathcal{L}(f_\Theta)\|_{L_1(\nu_n^X)}^2 \sum_{r=1}^m \left(\max_{j\in\{1,\dots,n\}}\frac{1}{m^\beta}|\sigma'(\theta_r^\top x_j)|\|x_j\|_2\right)^2 \\
&\leq \|\nabla_f \mathcal{L}(f_\Theta)\|_{L_1(\nu_n^X)}^2 m^{1-2\beta}K_1^2 \\
&\leq m^{1-2\beta}K_1^2.
\end{aligned}
\tag{19}
$$

Therefore, we get

$$
\begin{aligned}
\big|l(f_{\Theta+\tau}(x),y) &- \big(l(f_\Theta(x),y) - \eta\partial_\zeta l(f_\Theta(x),y)\nabla_\Theta f_\Theta(x)^\top\nabla_\Theta\mathcal{L}(\Theta)\big)\big| \\
&\leq \frac{1}{2}\left(\frac{K_1^2}{m^{2\beta-1}} + \frac{2K_2}{m^\beta} + \frac{\eta^2 K_1^2 K_2^2}{m^{4\beta-1}}\right)\eta^2\|\nabla_\Theta\mathcal{L}(\Theta)\|_2^2 \\
&\leq \frac{1}{2m^{2\beta-1}}\left(K_1^2 + 2K_2 + K_1^2 K_2^2\right)\eta^2\|\nabla_\Theta\mathcal{L}(\Theta)\|_2^2,
\end{aligned}
\tag{20}
$$

where we used $\beta\in[0,1)$ and $\eta\leq m^\beta$ for the last inequality.

Noting that from the definition of kernel smoothing of functional gradients (3), we see

$$
\begin{aligned}
\nabla_\Theta f_\Theta(x)^\top\nabla_\Theta\mathcal{L}(\Theta) &= \nabla_\Theta f_\Theta(x)^\top\left(\frac{1}{n}\sum_{i=1}^n \partial_\zeta l(f_\Theta(x_i),y_i)\nabla_\Theta f_\Theta(x_i)\right) \\
&= T_{k_\Theta}\nabla_f\mathcal{L}(f_\Theta)(x).
\end{aligned}
$$

Therefore, by taking the expectation of (20) according to the empirical distribution $\nu_n$, we get

$$
\begin{aligned}
\Big|\mathcal{L}(f_{\Theta+\tau}) &- \Big(\mathcal{L}(f_\Theta) - \eta\left\langle\nabla_f\mathcal{L}(f_\Theta),T_{k_\Theta}\nabla_f\mathcal{L}(f_\Theta)\right\rangle_{L_2(\nu_n^X)}\Big)\Big| \\
&\leq \frac{\eta^2}{2m^{2\beta-1}}\left(K_1^2 + 2K_2 + K_1^2 K_2^2\right)\|\nabla_\Theta\mathcal{L}(\Theta)\|_2^2.
\end{aligned}
$$

This completes the proof of the statement (i).

From the convexity of $l(\zeta,y)$ with respect to $\zeta$, we have

$$
\begin{aligned}
l(f_{\Theta^*}(X),Y) &= l\left(f_\Theta(X) + \nabla_\Theta f_\Theta(X)^\top(\Theta^* - \Theta) + g_X(\Theta^* - \Theta),Y\right) \\
&\geq l(f_\Theta(X),Y) + \partial_\zeta l(f_\Theta(X),Y)\left(\nabla_\Theta f_\Theta(X)^\top(\Theta^* - \Theta) + g_X(\Theta^* - \Theta)\right) \\
&\geq l(f_\Theta(X),Y) + \nabla_\Theta l(f_\Theta(X),Y)^\top(\Theta^* - \Theta) - |\partial_\zeta l(f_\Theta(X),Y)|\frac{K_2\|\Theta^* - \Theta\|_2^2}{m^\beta},
\end{aligned}
$$

where we used $|g_X(\Theta^* - \Theta)| \leq \frac{K_2\|\Theta^*-\Theta\|_2^2}{m^\beta}$. Thus, by taking the expectation with respect to $(X,Y)\sim\nu_n$, we get

$$
\mathcal{L}(\Theta^*) \geq \mathcal{L}(\Theta) + \nabla_\Theta\mathcal{L}(\Theta)^\top(\Theta^* - \Theta) - \frac{K_2}{m^\beta}\|\nabla_f\mathcal{L}(f_\Theta)\|_{L_1(\nu_n^X)}\|\Theta^* - \Theta\|_2^2.
$$

This finishes the proof of the statement (ii). $\qquad\square$

## D.2 PROOF OF PROPOSITION 6

*Proof of Proposition 6.* Set $Z_{r,i} \stackrel{def}{=} y_i \partial_\theta \sigma(\theta_r^{(0)} x_i)^\top v(\theta_r^{(0)})$. We find clearly $|Z_{r,i}| \leq K_1$ from Assumption 1. By applying Hoeffding's inequality to $Z_{r,i}$ for each $i \in \{1, \ldots, n\}$ and taking an union bound, we have

$$\mathbb{P}_{\Theta^{(0)}} \left[ \max_{i \in \{1,\ldots,n\}} \left| \frac{2}{m} \sum_{r=1}^{m/2} Z_{r,i} - \mathbb{E}_{\theta_r^{(0)}}[Z_{r,i}] \right| > \frac{\rho}{2} \right] \leq 2n \exp\left( -\frac{\rho^2 m}{16 K_1^2} \right).$$

In other words, since $m \geq \frac{16 K_1^2}{\rho^2} \log \frac{2n}{\delta}$, we have with probability $1 - \delta$,

$$\max_{i \in \{1,\ldots,n\}} \left| \frac{2}{m} \sum_{r=1}^{m/2} Z_{r,i} - \mathbb{E}_{\theta_r^{(0)}}[Z_{r,i}] \right| \leq \frac{\rho}{2}.$$

Therefore, using Assumption 1 **(A4)** and noting $\Theta^{(0)} = (\theta_r)_{r=1}^m$ is symmetrically initialized, we get with probability $1 - \delta$ for $\forall i \in \{1, \ldots, n\}$,

$$\frac{1}{m} \sum_{r=1}^m y_i \partial_\theta \sigma(\theta_r^{(0)} x_i)^\top v(\theta_r^{(0)}) \geq \frac{\rho}{2}. \tag{21}$$

In the following proof, we assume $\Theta^{(0)} = (\theta_r^{(0)})_{r=1}^m$ satisfies this inequality. We get from the $K_2$-Lipschitz continuity of $\sigma'$ that for $\Theta = (\theta_r)_{r=1}^m$ satisfying $\|\Theta - \Theta^{(0)}\|_{2,1} \leq \frac{m\rho}{4 K_2}$,

$$\left| \frac{1}{m} \sum_{r=1}^m y_i \sigma'(\theta_r^\top x_i) x_i^\top v(\theta_r^{(0)}) - \frac{1}{m} \sum_{r=1}^m y_i \sigma'(\theta_r^{(0)\top} x_i) x_i^\top v(\theta_r^{(0)}) \right|$$

$$\leq \frac{1}{m} \sum_{r=1}^m \left| y_i x_i^\top v_r(\theta_r^{(0)})(\sigma'(\theta_r^\top x_i) - \sigma'(\theta_r^{(0)\top} x_i)) \right|$$

$$\leq \frac{1}{m} \sum_{r=1}^m K_2 |(\theta_r - \theta_r^{(0)})^\top x_i|$$

$$\leq \frac{K_2}{m} \|\Theta - \Theta^{(0)}\|_{2,1}$$

$$\leq \frac{\rho}{4}.$$

This means that there exists $(v_r)_{r=1}^m$ such that $\|v_r\|_2 \leq 1$ ($\forall r \in \{1, \ldots, m\}$) and for $\forall \Theta = (\theta_r)_{r=1}^m$ satisfying $\|\Theta - \Theta^{(0)}\|_{2,1} \leq \frac{m\rho}{4 K_2}$,

$$\frac{1}{m} \sum_{r=1}^m y_i \partial_\theta \sigma(\theta_r^\top x_i)^\top v_r \geq \frac{\rho}{4}, \quad \forall i \in \{1, \ldots, n\}.$$

Then, we get the following bound: for $\forall (\alpha_i)_{i=1}^n$ ($\alpha_i \in (0,1)$),

$$\frac{1}{m} \sum_{i=1}^n \sum_{r=1}^m y_i \alpha_i \partial_\theta \sigma(\theta_r^\top x_i)^\top v_r = \frac{1}{m} \sum_{i=1}^n \alpha_i \sum_{r=1}^m y_i \partial_\theta \sigma(\theta_r^\top x_i)^\top v_r \geq \frac{\rho}{4} \sum_{i=1}^n \alpha_i > 0. \tag{22}$$

Noting that $\nabla_f \mathcal{L}(f_\Theta)(x_i) = \frac{-y_i}{1+\exp(y_i f_\Theta(x_i))}$, we get

$$
\begin{aligned}
\langle \nabla_f \mathcal{L}(f_\Theta), T_{k_\Theta} \nabla_f \mathcal{L}(f_\Theta) \rangle_{L_2(\nu_n^X)} &= \frac{1}{n^2} \sum_{i,j=1}^n k_\Theta(x_i, x_j) \nabla_f \mathcal{L}(f_\Theta)(x_i) \nabla_f \mathcal{L}(f_\Theta)(x_j) \\
&= \frac{1}{n^2} \left\| \sum_{i=1}^n \nabla_f \mathcal{L}(f_\Theta)(x_i) \nabla_\Theta f_\Theta(x_i) \right\|_2^2 \\
&= \frac{1}{n^2} \sum_{r=1}^m \left\| \sum_{i=1}^n \nabla_f \mathcal{L}(f_\Theta)(x_i) \partial_{\theta_r} f_\Theta(x_i) \right\|_2^2 \\
&= \frac{1}{n^2} \sum_{r=1}^m \left\| \frac{1}{m^\beta} \sum_{i=1}^n \nabla_f \mathcal{L}(f_\Theta)(x_i) \partial_\theta \sigma(\theta_r^\top x_i) \right\|_2^2 \\
&\geq \frac{1}{n^2} \sum_{r=1}^m \left( \frac{1}{m^\beta} \sum_{i=1}^n \nabla_f \mathcal{L}(f_\Theta)(x_i) \partial_\theta \sigma(\theta_r^\top x_i)^\top v_r \right)^2 \\
&\geq \frac{m}{n^2} \left( \frac{1}{m^{1+\beta}} \sum_{i=1}^n \sum_{r=1}^m \nabla_f \mathcal{L}(f_\Theta)(x_i) \partial_\theta \sigma(\theta_r^\top x_i)^\top v_r \right)^2 \\
&\geq \frac{m^{1-2\beta} \rho^2}{16 n^2} \left( \sum_{i=1}^n \frac{1}{1+\exp(y_i f_\Theta(x_i))} \right)^2,
\end{aligned}
$$

where we used $\|v_r\|_2 \leq 1$ for the first inequality, the convexity of $\|\cdot\|_2^2$ for the second inequality, and (22) for the last inequality. We can find that this inequality finishes the proof because

$$
\frac{1}{1+\exp(f_\Theta(x_i)y_i)} = \frac{1}{2} |y_i - 2p_\Theta(Y=1 \mid x_i) + 1|.
$$

$\square$

### D.3 PROOF OF PROPOSITION 7

The proof of Proposition 7 is based on the traditional convergence analysis of gradient descent for smooth objective functions in finite-dimensional space.

*Proof of Proposition 7.* We first specify the smoothness of the logistic loss function. We set $\phi(v) = \log(1+\exp(-v))$ and $l(y, f_\Theta(x)) = \phi(y f_\Theta(x))$. By the simple calculation, we get that for $r, s \in \{1, \ldots, m\}$,

$$
\begin{aligned}
\frac{\partial^2}{\partial \theta_r \partial \theta_s} l(y, f_\Theta(x)) &= \phi''(y f_\Theta(x)) \frac{a_r a_s}{m^{2\beta}} \sigma'(\theta_r^\top x) \sigma'(\theta_s^\top x) x x^\top \\
&\quad + \mathbf{1}[r=s] \frac{y}{m^\beta} \phi'(y f_\Theta(x)) a_r \sigma''(\theta_r^\top x) x x^\top.
\end{aligned}
$$

Noting that $\|\phi'\|_\infty \leq 1$ and $\|\phi''\|_\infty \leq \frac{1}{4}$, we can see that the maximum eigen-value of $(\partial^2 l(y, f_\Theta(x))/\partial \theta_r \partial \theta_s)_{r,s=1}^m$ is upper bounded by

$$
M \overset{def}{=} \frac{1}{4 m^{2\beta-1}} (K_1^2 + K_2).
$$

Indeed, for $v = (v_r)_{r=1}^m$ such that $\sum_{r=1}^m \|v_r\|_2^2 \le 1$, $(v_r \in \mathbb{R}^d)$, we have

$$\sum_{r,s=1}^m v_r^\top \frac{\partial^2 l(y, f_\Theta(x))}{\partial \theta_r \partial \theta_s} v_s = \frac{\phi''(y f_\Theta(x))}{m^{2\beta}} \left( \sum_{r=1}^m a_r \sigma'(\theta_r^\top x) v_r^\top x \right)^2 + \frac{y}{m^\beta} \phi'(y f_\Theta(x)) \sum_{r=1}^m a_r \sigma''(\theta_r^\top x)(v_r^\top x)^2$$

$$\le \frac{K_1^2}{4m^{2\beta}} \left( \sum_{r=1}^m \|v_r\|_2 \right)^2 + \frac{K_2}{m^\beta} \sum_{r=1}^m \|v_r\|_2^2$$

$$\le \frac{K_1^2}{4m^{2\beta}} \left( \sqrt{m} \|v\|_2 \right)^2 + \frac{K_2}{m^\beta}$$

$$\le \frac{K_1^2}{4m^{2\beta-1}} + \frac{K_2}{m^\beta}$$

$$\le \frac{1}{4m^{2\beta-1}} (K_1^2 + K_2).$$

Therefore, the loss function $\mathcal{L}(\Theta)$ is $M$-Lipschitz smooth with respect to $\Theta$, that is, for

$$\mathcal{L}(\Theta') \le \mathcal{L}(\Theta) + \langle \nabla \mathcal{L}(\Theta), \Theta' - \Theta \rangle_2 + \frac{M}{2} \|\Theta' - \Theta\|_2^2.$$

Plugging $\Theta = \Theta^{(t)}$ and $\Theta' = \Theta^{(t+1)} = \Theta - \eta \nabla_\Theta \mathcal{L}(\Theta^{(t)})$ into this inequality, we get

$$\mathcal{L}(\Theta^{(t+1)}) \le \mathcal{L}(\Theta^{(t)}) - \eta \left( 1 - \frac{\eta M}{2} \right) \|\nabla \mathcal{L}(\Theta^{(t)})\|_2^2$$

$$\le \mathcal{L}(\Theta^{(t)}) - \frac{\eta}{2} \|\nabla \mathcal{L}(\Theta^{(t)})\|_2^2,$$

where we used $\eta \le 1/M$ for the last inequality. By summing this inequality over $t \in \{0, \ldots, T-1\}$ and multiplying by $\frac{2}{\eta T}$, we get

$$\frac{1}{T} \sum_{t=0}^{T-1} \|\nabla \mathcal{L}(\Theta^{(t)})\|_2^2 \le \frac{2}{\eta T} \mathcal{L}(\Theta^{(0)}) = \frac{2}{\eta T} \log(2), \tag{23}$$

where we used $\mathcal{L}(\Theta^{(0)}) = \log(2)$. Therefore, we have that from equation (23),

$$\|\Theta^{(t)} - \Theta^{(0)}\|_2 \le \eta \sum_{t=0}^{T-1} \left\| \nabla_\Theta \mathcal{L}(\Theta^{(t)}) \right\|_2$$

$$\le \eta \sqrt{T} \sqrt{\sum_{t=0}^{T-1} \left\| \nabla_\Theta \mathcal{L}(\Theta^{(t)}) \right\|_2^2}$$

$$\le \sqrt{2\eta T \log(2)}.$$

The last statement of Proposition 7 immediately follows from this and the following inequality.

$$\|\Theta^{(t)} - \Theta^{(0)}\|_{2,1} = \sum_{r=1}^m \|\theta_r^{(t)} - \theta_r^{(0)}\|_2$$

$$\le \sqrt{m} \sqrt{\sum_{r=1}^m \|\theta_r^{(t)} - \theta_r^{(0)}\|_2^2}$$

$$= \sqrt{m} \|\Theta^{(t)} - \Theta^{(0)}\|_2.$$

$\square$

# E    PROOFS FOR GENERALIZATION BOUNDS

*Proof of Proposition 8.* In this proof, we denote $\mathcal{F} = \mathcal{F}_{\eta,m,T}^\gamma$ and $\Omega = \Omega_{\eta,m,T}$ for simplicity. We define $\mathcal{F}_y \overset{def}{=} \{h(\cdot, y) : \mathcal{X} \to [0,1] \mid h \in \mathcal{F}\}$. Then, for a given dataset $S = (x_i, y_i)_{i=1}^n$, we notice

that $\Re(\mathcal{F}|_S) \le \Re(\mathcal{F}_1|_X) + \Re(\mathcal{F}_{-1}|_X)$, where $X = (x_i)_{i=1}^n$. Thus, it is enough to provide an upper bound on $\Re(\mathcal{F}_1|_X)$ because a bound on the other complexity can be also derived in the same way.

We first give a uniform high probability bound on the initialization $\|\theta_r^{(0)}\|_2$ for $\forall r \in \{1, \dots, m\}$. We get from **(A2)**, for $t > 0$,

$$\mathbb{P}\left[\max_{r \in \{1,\dots,m\}} \|\theta_r^{(0)}\|_2 \ge t\right] \le \sum_{r=1}^m \mathbb{P}\left[\|\theta_r^{(0)}\|_2 \ge t\right] \le mA \exp(-bt^2).$$

Thus, by choosing $t$ so that $\delta = mA \exp(-bt^2)$, we confirm that with probability at least $1 - \delta$,

$$\max_{r \in \{1,\dots,m\}} \|\theta_r^{(0)}\|_2 \le R \overset{def}{=} \sqrt{\frac{1}{b} \log\left(\frac{mA}{\delta}\right)}.$$

We introduce several notations. Fix $R_0 > 0$. We denote $\overline{\theta} = (\theta, \theta') \in \mathbb{R}^{2d}$, $(\theta, \theta' \in \mathbb{R}^d)$ and, define for $\overline{\theta}$,

$$g_{\overline{\theta}}(x) \overset{def}{=} \frac{\sigma(\theta^\top x) - \sigma(\theta'^\top x)}{\|\theta - \theta'\|_2}.$$

When $\theta = \theta'$, we define $g_{\overline{\theta}}(x) = 0$. From the Lipschitz continuity of $\sigma$, the range of $g_{\overline{\theta}}$ is $[-K_1, K_1]$. Moreover, we define

$$\overline{\Omega} \overset{def}{=} \{\overline{\theta} \in \mathbb{R}^{2d} \mid \|\theta\|_2, \|\theta'\|_2 \le R + D_{\eta,m,T}, \; \|\theta - \theta'\|_2 \le D_{\eta,m,T}\},$$

$$\overline{\Omega}_+ \overset{def}{=} \{\overline{\theta} \in \overline{\Omega} \mid R_0 < \|\theta - \theta'\|_2 \le D_{\eta,m,T}\},$$

$$\overline{\Omega}_- \overset{def}{=} \{\overline{\theta} \in \overline{\Omega} \mid \|\theta - \theta'\|_2 \le R_0\},$$

$$\mathcal{G}_+ \overset{def}{=} \left\{g_{\overline{\theta}} : \mathcal{X} \to [-K_1, K_1] \mid \overline{\theta} \in \overline{\Omega}_+\right\},$$

$$\mathcal{G}_- \overset{def}{=} \left\{g_{\overline{\theta}} : \mathcal{X} \to [-K_1, K_1] \mid \overline{\theta} \in \overline{\Omega}_-\right\},$$

$$\mathcal{H} \overset{def}{=} \{f_\Theta : \mathcal{X} \to \mathbb{R} \mid \Theta \in \Omega\}.$$

Clearly, we see

$$\overline{\Omega} = \overline{\Omega}_- \cup \overline{\Omega}_+ \text{ and } \left\{g_{\overline{\theta}} \mid \overline{\theta} \in \overline{\Omega}\right\} = \mathcal{G}_- \cup \mathcal{G}_+.$$

From the Lipschitz continuity of $l_\gamma$, we find $\Re(\mathcal{F}_1|_X) \le \gamma^{-1} \Re(\mathcal{H}|_X)$.

We now derive an upper bound on the Rademacher complexity. Set $C_M \overset{def}{=} m^{\frac{1}{2}-\beta} D_{\eta,m,T}$.

$$\begin{aligned}
\Re(\mathcal{H}|_X) &= \frac{1}{n} \mathbb{E}\left[\sup_{\Theta \in \Omega} \sum_{i=1}^n \epsilon_i f_\Theta(x_i)\right] \\
&= \frac{1}{n} \mathbb{E}\left[\sup_{\Theta \in \Omega} \sum_{i=1}^n \epsilon_i (f_\Theta(x_i) - f_{\Theta^{(0)}}(x_i))\right] \\
&= \frac{1}{n} \mathbb{E}\left[\sup_{\Theta \in \Omega} \sum_{i=1}^n \epsilon_i \frac{1}{m^\beta} \sum_{r=1}^m a_r \left(\sigma(\theta_r^\top x_i) - \sigma(\theta_r^{(0)\top} x_i)\right)\right] \\
&= \frac{C_M}{n} \mathbb{E}\left[\sup_{\Theta \in \Omega} \sum_{i=1}^n \epsilon_i \sum_{r=1}^m \frac{\|\theta_r - \theta_r^{(0)}\|_2}{C_M m^\beta} a_r \frac{\sigma(\theta_r^\top x_i) - \sigma(\theta_r^{(0)\top} x_i)}{\|\theta_r - \theta_r^{(0)}\|_2}\right],
\end{aligned}$$

(24)

where we used the fact that $f_{\Theta^{(0)}}(x_i)$ is a constant in the expectation for the second equality.

Since, for $\Theta \in \Omega$,

$$\sum_{r=1}^m \frac{\|\theta_r - \theta_r^{(0)}\|_2}{C_M m^\beta} \le \frac{m^{\frac{1}{2}-\beta} \|\Theta - \Theta^{(0)}\|_2}{C_M} \le 1,$$

equation (24) can be upper-bounded by the Rademacher complexity of the convex hull. Hence,

$$\Re(\mathcal{H}|_X) \leq \frac{C_M}{n} \mathbb{E}\left[\sup_{\substack{\Theta \in \Omega \\ \sum_{r=1}^m b_r \leq 1, b_r \in [0,1]}} \sum_{i=1}^n \epsilon_i \sum_{r=1}^m b_r a_r \frac{\sigma(\theta_r^\top x_i) - \sigma(\theta_r^{(0)\top} x_i)}{\|\theta_r - \theta_r^{(0)}\|_2}\right]$$

$$\leq \frac{C_M}{n} \mathbb{E}\left[\sup_{\substack{(\theta_r, \theta_r')_{r=1}^m \in \overline{\Omega}^m \\ \sum_{r=1}^m b_r \leq 1, b_r \in [0,1]}} \sum_{i=1}^n \epsilon_i \sum_{r=1}^m b_r \frac{\sigma(\theta_r^\top x_i) - \sigma(\theta_r'^\top x_i)}{\|\theta_r - \theta_r'\|_2}\right]$$

$$= \frac{C_M}{n} \mathbb{E}\left[\sup_{(\theta, \theta') \in \overline{\Omega}} \sum_{i=1}^n \epsilon_i \frac{\sigma(\theta^\top x_i) - \sigma(\theta'^\top x_i)}{\|\theta - \theta'\|_2}\right]$$

$$= \frac{C_M}{n} \mathbb{E}\left[\sup_{\overline{\theta} \in \overline{\Omega}} \sum_{i=1}^n \epsilon_i g_{\overline{\theta}}(x_i)\right]$$

$$\leq \frac{C_M}{n} \mathbb{E}\left[\sup_{g \in \mathcal{G}_-} \sum_{i=1}^n \epsilon_i g(x_i) + \sup_{g \in \mathcal{G}_+} \sum_{i=1}^n \epsilon_i g(x_i)\right] = C_M \left(\Re(\mathcal{G}_-|_X) + \Re(\mathcal{G}_+|_X)\right). \quad (25)$$

We used that for $\Theta \in \Omega$, $(\theta_r, \theta_r^{(0)}) \in \overline{\Omega}$ $(\forall r \in \{1, \ldots, m\})$ because $\|\Theta - \Theta^{(0)}\|_2 \leq D_{\eta,m,T}$. Moreover, the term $a_r$ disappeared by the symmetry. We used the fact that the convex hull of a hypothesis class does not increase the Rademacher complexity for the first equality.

We next derive an upper bound on the Rademacher complexity $\Re(\mathcal{G}_+|_X)$ through the covering number $\mathcal{N}(\mathcal{G}_+|_X, \epsilon, \|\cdot\|_\infty)$ and Lemma 2. To this end, we investigate the sensitivity of $\|g_{\overline{\theta}}\|_\infty$ with respect to $\overline{\theta}$ as follows.

Let $\overline{\theta}_1 = (\theta_1, \theta_1') \in \overline{\Omega}_+$ and $\overline{\theta}_2 = (\theta_2, \theta_2') \in \overline{\Omega}_+$ be parameters such that

$$\|\overline{\theta}_1 - \overline{\theta}_2\|_2 = \sqrt{\|\theta_1 - \theta_2\|_2^2 + \|\theta_1' - \theta_2'\|_2^2} \leq \epsilon.$$

This leads to

$$\|\theta_1 - \theta_2\|_2, \|\theta_1' - \theta_2'\|_2 \leq \epsilon \text{ and } |\|\theta_1 - \theta_1'\|_2 - \|\theta_2 - \theta_2'\|_2| \leq 2\epsilon.$$

We get from these inequalities that for $\|x\|_2 \leq 1$,

$$|g_{\overline{\theta}_1}(x) - g_{\overline{\theta}_2}(x)| = \frac{\left|\|\theta_2 - \theta_2'\|_2 (\sigma(\theta_1^\top x) - \sigma(\theta_1'^\top x)) - \|\theta_1 - \theta_1'\|_2 (\sigma(\theta_2^\top x) - \sigma(\theta_2'^\top x))\right|}{\|\theta_1 - \theta_1'\|_2 \|\theta_2 - \theta_2'\|_2}$$

$$\leq \frac{\left|(\|\theta_2 - \theta_2'\|_2 - \|\theta_1 - \theta_1'\|_2)(\sigma(\theta_1^\top x) - \sigma(\theta_1'^\top x))\right|}{\|\theta_1 - \theta_1'\|_2 \|\theta_2 - \theta_2'\|_2}$$

$$+ \frac{\left|\|\theta_1 - \theta_1'\|_2 (\sigma(\theta_1^\top x) - \sigma(\theta_1'^\top x) - \sigma(\theta_2^\top x) + \sigma(\theta_2'^\top x))\right|}{\|\theta_1 - \theta_1'\|_2 \|\theta_2 - \theta_2'\|_2}$$

$$\leq \frac{4 D_{\eta,m,T} \epsilon K_1}{R_0^2}.$$

Thus, if $\|\overline{\theta}_1 - \overline{\theta}_2\|_2 \leq \epsilon$ for $\overline{\theta}_1, \overline{\theta}_2 \in \overline{\Omega}_+$, then $\|g_{\overline{\theta}_1} - g_{\overline{\theta}_2}\|_\infty \leq 4 D_{\eta,m,T} \epsilon K_1 / R_0^2$. Since,

$$\mathcal{G}_+ \subset \{g_{\overline{\theta}} \mid \|\overline{\theta}\|_2 \leq 2R + 2D_{\eta,m,T}, \ \overline{\theta} \in \mathbb{R}^{2d}\},$$

we get for the unit-ball $B_1 \subset \mathbb{R}^{2d}$ with respect to $\|\cdot\|_2$,

$$\mathcal{N}(\mathcal{G}_+|_X, \epsilon, \|\cdot\|_\infty) \leq C_1^d \mathcal{N}\left(B_1, \frac{R_0^2 \epsilon}{K_1(RD_{\eta,m,T} + D_{\eta,m,T}^2)}, \|\cdot\|_2\right),$$

where $C_1 > 0$ is a uniform constant. Hence,

$$\log \mathcal{N}(\mathcal{G}_+|_X, \epsilon, \|\cdot\|_\infty) \leq O\left(d \log\left(1 + \frac{K_1(RD_{\eta,m,T} + D_{\eta,m,T}^2)}{R_0^2 \epsilon}\right)\right).$$

Applying Lemma 2 with $\alpha = K_1/\sqrt{n}$, we obtain

$$\Re(\mathcal{G}_+|_X) = O\left(K_1\sqrt{\frac{d}{n}\log\left(1 + \frac{\sqrt{n}(RD_{\eta,m,T} + D_{\eta,m,T}^2)}{R_0^2}\right)}\right). \tag{26}$$

We next evaluate $\Re(\mathcal{G}_-|_X)$ by using a linear approximation. Since $|\sigma''(\cdot)| \leq K_2$, we get

$$|\sigma({\theta'}^\top x) - \sigma(\theta^\top x) - \sigma'(\theta^\top x)(\theta' - \theta)^\top x| \leq K_2\|\theta' - \theta\|_2^2.$$

Therefore, we get for $\overline{\theta} = (\theta, \theta') \in \overline{\Omega}_-$,

$$\left|g_{\overline{\theta}}(x) - \frac{\sigma'(\theta^\top x)(\theta' - \theta)^\top x}{\|\theta - \theta'\|_2}\right| \leq K_2\|\theta - \theta'\|_2 \leq K_2 R_0.$$

From this approximation, the Rademacher complexity can be bounded as follows.

$$\Re(\mathcal{G}_-|_X) \leq K_2 R_0 + \frac{1}{n}\mathbb{E}\left[\sup_{\overline{\theta}\in\overline{\Omega}_-}\sum_{i=1}^n \epsilon_i \frac{\sigma'(\theta^\top x_i)(\theta' - \theta)^\top x_i}{\|\theta - \theta'\|_2}\right]$$

$$\leq K_2 R_0 + \frac{1}{n}\mathbb{E}\left[\sup_{\substack{\|\theta\|_2 \leq R + D_{\eta,m,T}, \\ \|w\|_2 \leq 1}}\sum_{i=1}^n \epsilon_i \sigma'(\theta^\top x_i)w^\top x_i\right].$$

When $\sqrt{\|\theta_1 - \theta_2\|_2^2 + \|w_1 - w_2\|_2^2} \leq \epsilon$ for $\|\theta_i\|_2 \leq R + D_{\eta,m,T}$ and $\|w_i\|_2 \leq 1$, we get for $\|x\|_2 \leq 1$,

$$|\sigma'(\theta_1^\top x)w_1^\top x - \sigma'(\theta_2^\top x)w_2^\top x| \leq |(\sigma'(\theta_1^\top x) - \sigma'(\theta_2^\top x))w_1^\top x| + |\sigma'(\theta_2^\top x)(w_1 - w_2)^\top x|$$
$$\leq (K_1 + K_2)\epsilon.$$

We set

$$\mathcal{G}'_- \overset{def}{=} \{x \to \sigma'(\theta^\top x)w^\top x \mid \|\theta\|_2 \leq R + D_{\eta,m,T}, \|w\|_2 \leq 1\}.$$

Therefore, by the same argument as the case of $\mathcal{G}_+$, the following bound holds.

$$\mathcal{N}(\mathcal{G}'_-|_X, \epsilon, \|\cdot\|_\infty) \leq C_2^d \mathcal{N}\left(B_1, \frac{\epsilon}{(K_1 + K_2)(R + D_{\eta,m,T})}, \|\cdot\|_2\right),$$

where $C_2 > 0$ is a uniform constant. Hence,

$$\log\mathcal{N}(\mathcal{G}'_-|_X, \epsilon, \|\cdot\|_\infty) \leq O\left(d\log\left(1 + \frac{(K_1 + K_2)(R + D_{\eta,m,T})}{\epsilon}\right)\right).$$

By Lemma 2 with $\alpha = 1/\sqrt{n}$, we get

$$\Re(\mathcal{G}_-|_X) \leq O\left(K_2 R_0 + \sqrt{\frac{d}{n}\log\left(1 + \sqrt{n}(K_1 + K_2)(R + D_{\eta,m,T})\right)}\right). \tag{27}$$

Combining (25), (26), (27) with $R_0 = \sqrt{d/n}$, and Lipschiz continuity of $l_\gamma$, we obtain

$$\Re(\mathcal{F}_1|_X) \leq O\left(\frac{m^{\frac{1}{2}-\beta}D_{\eta,m,T}}{\gamma}(1 + K_1 + K_2)\sqrt{\frac{d}{n}\log\left(n(1 + K_1 + K_2)(R + D_{\eta,m,T})\right)}\right).$$

Now, let us turn to the second part of Proposition 8. Let us assume that the function $\sigma$ is convex and satisfies $\sigma(0) = 0$. The main argument uses the convexity of activation function in the same spirit as (Chinot et al., 2019). As for the first part, with probability larger than $1 - \delta$ over the initialization

$$\max_{r\in\{1,\ldots,m\}}\|\theta_r^{(0)}\|_2 \leq R \overset{def}{=} \sqrt{\frac{1}{b}\log\left(\frac{mA}{\delta}\right)}.$$

We only focus on the control of $\Re(\mathcal{H}|_X)$.

$$
\begin{aligned}
\Re(\mathcal{H}|_X) &= \frac{1}{n}\mathbb{E}\left[\sup_{\Theta\in\Omega}\sum_{i=1}^{n}\epsilon_i f_\Theta(x_i)\right] \\
&= \frac{1}{n}\mathbb{E}\left[\sup_{\Theta\in\Omega}\sum_{i=1}^{n}\epsilon_i\big(f_\Theta(x_i)-f_{\Theta^{(0)}}(x_i)\big)\right] \\
&= \frac{1}{n}\mathbb{E}\left[\sup_{\Theta\in\Omega}\sum_{i=1}^{n}\epsilon_i\sum_{r=1}^{m}\frac{a_r}{m^\beta}\big(\sigma(\theta_r^T x_i)-\sigma(\theta_r^{(0)T}x_i)\big)\right] \\
&= \frac{1}{n}\mathbb{E}\left[\sup_{\Theta\in\Omega}\sum_{(i,r)\in\mathcal{A}}\epsilon_i\frac{a_r}{m^\beta}\big(\sigma(\theta_r^T x_i)-\sigma(\theta_r^{(0)T}x_i)\big)\right] \\
&\quad + \frac{1}{n}\mathbb{E}\left[\sup_{\Theta\in\Omega}\sum_{(i,r)\in\mathcal{A}^c}\epsilon_i\frac{a_r}{m^\beta}\big(\sigma(\theta_r^{(0)T}x_i)-\sigma(\theta_r^T x_i)\big)\right]
\end{aligned}
$$

where $\mathcal{A}=\{(i,r)\in\{1,\cdots,n\}\times\{1,\cdots,m\}:\sigma(\theta_r^T x_i)-\sigma(\theta_r^{(0)T}x_i)\geq 0\}$.

Let us control the first term (i.e for $(i,r)\in\mathcal{A}$). For any $i,r$ in $\mathcal{A}$ let $\psi_{i,r}:\mathbb{R}\mapsto\mathbb{R}$ defined for all $u\in\mathbb{R}$ as:

$$
\psi_{i,r}(u)=\sigma(u+\theta_r^{(0)T}x_i)-\sigma(\theta_r^{(0)T}x_i)
$$

The functions $\psi_{i,r}$ are such that $\psi_{i,r}(0)=0$. There are convex because $\sigma$ is. In particular for any $\alpha\geq 1$ and $u\in\mathbb{R}$, $\psi_{i,r}(\alpha u)\geq\alpha\psi_{i,r}(u)$. We also have $\psi_{i,r}\big((\theta_r-\theta_r^{(0)})^T x_i\big)=\sigma(\theta_r^T x_i)-\sigma(\theta_r^{(0)T}x_i)$. Since $\Theta\in\Omega$ we have $\|\Theta-\Theta^{(0)}\|_2\leq D_{\eta,m,T}$ and for any $r\in\{1,\cdots,m\}$, $\|\theta_r-\theta_r^{(0)}\|_2\leq D_{\eta,m,T}$. As a consequence, for any $(i,r)\in\mathcal{A}$, there exists $\beta_{i,r}\in[0,1]$ such that

$$
\frac{D_{\eta,m,T}}{\|\theta_r-\theta_r^{(0)}\|_2}\psi_{i,r}\big((\theta_r-\theta_r^{(0)})^T x_i\big)=\beta_{i,r}\psi_{i,r}\left(\frac{D_{\eta,m,T}}{\|\theta_r-\theta_r^{(0)}\|_2}(\theta_r-\theta_r^{(0)})^T x_i\right)
$$

Since for any $i\in\{1,\cdots,n\}$, $\sum_{r=1}^{m}\frac{\|\theta_r-\theta_r^{(0)}\|_2}{C_M m^\beta}\beta_{i,r}\leq 1$, we get

$$
\frac{1}{n}\mathbb{E}\sup_{\Theta\in\Omega}\sum_{(i,r)\in\mathcal{A}}\epsilon_i\frac{a_r}{m^\beta}\big(\sigma(\theta_r^T x_i)-\sigma(\theta_r^{(0)T}x_i)\big)
$$

$$
\leq\frac{1}{n}\frac{C_M}{D_{\eta,m,T}}\mathbb{E}\left[\sup_{\Theta=(\theta_r)_{r=1}^m:\|\theta_r-\theta_r^{(0)}\|_2\leq D_{\eta,m,T}}\sum_{(i,j)\in\mathcal{A}}\epsilon_i a_r\frac{\|\theta_r-\theta_r^{(0)}\|_2}{C_M m^\beta}\beta_{i,r}\psi_{i,r}\left(\frac{D_{\eta,m,T}(\theta_r-\theta_r^{(0)})^T x_i}{\|\theta_r-\theta_r^{(0)}\|_2}\right)\right]
$$

$$
\leq\frac{1}{n}\frac{C_M}{D_{\eta,m,T}}\mathbb{E}\left[\sup_{\substack{\Theta=(\theta_r)_{r=1}^m:\|\theta_r\|_2\leq D_{\eta,m,T}\\ b=(b_r)_{r=1}^m,b_r\in[0,1],\sum_{r=1}^m b_r\leq 1}}\sum_{i=1}^{n}\epsilon_i\sum_{r=1}^{m}a_r b_r\psi_{i,r}\big(\theta_r^T x_i\big)\right]
$$

$$
=\frac{1}{n}\frac{C_M}{D_{\eta,m,T}}\mathbb{E}\left[\sup_{\substack{\Theta=(\theta_r)_{r=1}^m:\|\theta_r\|_2\leq D_{\eta,m,T}\\ b=(b_r)_{r=1}^m,b_r\in[0,1],\sum_{r=1}^m b_r\leq 1}}\sum_{i=1}^{n}\epsilon_i\sum_{r=1}^{m}a_r b_r\big(\sigma((\theta_r+\theta_r^{(0)})^T x_i)-\sigma(\theta_r^{(0)T}x_i)\big)\right]\quad.
$$

Therefore, with probability larger than $1 - \delta$,

$$\frac{1}{n} \mathbb{E} \sup_{\Theta \in \Omega} \sum_{(i,r) \in \mathcal{A}} \epsilon_i \frac{a_r}{m^\beta} \big( \sigma(\theta_r^T x_i) - \sigma(\theta_r^{(0)T} x_i) \big)$$

$$\leq \frac{1}{n} \frac{C_M}{D_{\eta,m,T}} \mathbb{E} \left[ \sup_{\substack{\|\theta_r\|_2 \leq D_{\eta,m,T}; \|\tilde{\theta}_r\|_2 \leq R \\ b = (b_r)_{r=1}^m, b_r \in [0,1], \sum_{r=1}^m b_r \leq 1}} \sum_{i=1}^n \epsilon_i \sum_{r=1}^m a_r b_r \big( \sigma((\theta_r + \tilde{\theta}_r)^T x_i) - \sigma(\tilde{\theta}_r^T x_i) \big) \right]$$

$$\leq \frac{1}{n} \frac{C_M}{D_{\eta,m,T}} \mathbb{E} \left[ \sup_{\theta: \|\theta\|_2 \leq D_{\eta,m,T}; \|\tilde{\theta}_r\|_2 \leq R} \sum_{i=1}^n \epsilon_i \big( \sigma((\theta + \tilde{\theta})^T x_i) + \sup_{\|\tilde{\theta}_r\| \leq R} \sum_{i=1}^n \epsilon_i \sigma(\tilde{\theta}^T x_i) \big) \right]$$

$$\leq \frac{K_1}{n} \frac{C_M}{D_{\eta,m,T}} \mathbb{E} \left[ \sup_{\theta: \|\theta\|_2 \leq D_{\eta,m,T}; \|\tilde{\theta}_r\|_2 \leq R} \sum_{i=1}^n \epsilon_i (\theta + \tilde{\theta})^T x_i + \sup_{\|\tilde{\theta}_r\|_2 \leq R} \sum_{i=1}^n \epsilon_i \tilde{\theta}^T x_i \right]$$

$$\leq \frac{K_1}{\sqrt{n}} \frac{C_M}{D_{\eta,m,T}} (D_{\eta,m,T} + 2R) = \frac{K_1 m^{1/2 - \beta} (D_{\eta,m,T} + 2R)}{\sqrt{n}} \ .$$

Let us turn to the second term. For any $(i, r)$ in $\mathcal{A}^c$ let $\tilde{\psi}_{i,r} : \mathbb{R} \mapsto \mathbb{R}$ defined for all $u \in \mathbb{R}$ as:

$$\tilde{\psi}_{i,r}(u) = \sigma(u + \theta_r^T x_i) - \sigma(\theta_r^T x_i)$$

We have $\tilde{\psi}_{i,r}\big((\theta_r^{(0)} - \theta_r)^T x_i\big) = \sigma(\theta_r^{(0)T} x_i) - \sigma(\theta_r^T x_i)$. Using the same path as for $(i, j) \in \mathcal{A}$, with probability larger than $1 - \delta$, we obtain,

$$\frac{1}{n} \mathbb{E} \sup_{\Theta \in \Omega} \sum_{(i,r) \in \mathcal{A}^c} \epsilon_i \frac{a_r}{m^\beta} \big( \sigma(\theta_r^{(0)T} x_i) - \sigma(\theta_r^T x_i) \big)$$

$$\leq \frac{1}{n} \frac{C_M}{D_{\eta,m,T}} \mathbb{E} \left[ \sup_{\substack{\|\theta_r\|_2 \leq D_{\eta,m,T}, \|\tilde{\theta}_r\|_2 \leq R + D_{\eta,m,T} \\ b = (b_r)_{r=1}^m, b_r \in [0,1], \sum_{r=1}^m b_r \leq 1}} \sum_{i=1}^n \epsilon_i \sum_{r=1}^m a_r b_r \big( \sigma((\theta_r + \tilde{\theta}_r)^T x_i) - \sigma(\tilde{\theta}_r^T x_i) \big) \right] \ .$$

$$\leq \frac{K_1}{\sqrt{n}} \frac{C_M}{D_{\eta,m,T}} (3 D_{\eta,m,T} + 2R) \ .$$

$\square$

