# OpenReview forum: "Gradient Descent can Learn Less Over-parameterized Two-layer Neural Networks on Classification Problems"
_ICLR.cc/2020/Conference — Reject_

### Official Review · AnonReviewer3 · 2019-10-22
**Official Blind Review #3**

**Rating:** 8

**Review:**

The authors study the problem of binary logistic regression in a two-layer network with a smooth activation function.  They introduce a separability assumption on the dataset using the neural tangent model.  This separability assumption is weaker than the more Neural Tangent Kernel assumption that has been extensively studied in the regression literature.  In that case, a certain Gram-matrix must be nonnegative.  In the current work, the authors observe that the structure of the logistic loss in the binary classification problem restricts the functional gradients to lie in a particular space, meaning that nonnegative of the Gram-matrix is only needed on a subspace.  This is the underlying theoretical reason for why they can get improvement over those methods in the setting they study.  Under the separability assumption, the authors prove convergent gradient descent and generalization of the ensuring net, while assuming the two-layer networks are less overparameterized than what would have been possible under the Gram-matrix perspective.

This paper appears to be a significant contribution to the field of convergent gradient descent algorithms because of the introduction of a weaker condition that guarantees convergence.  While the work only applies to smooth activations and to logistic loss classification problems, it can inspire additional work both in rigorous guarantees for training neural nets in regression and classification.  As a result, I recommend the paper be accepted for ICLR.

Minor comments:

(1) The abstract, title, and introduction emphasize the aspect of being "less overparameterized" than other methods.  It would be helpful to readers to have an absolute claim instead of a relative claim.
(2) The abstract claims the separability assumption is "more reasonable" than the positivity condition.  This claim is overly vague and should be clarified.
(3) There is a stray \forall in the third line of Theorem 2.

**Experience Assessment:**

I do not know much about this area.

**Review Assessment: Checking Correctness Of Derivations And Theory:**

I did not assess the derivations or theory.

**Review Assessment: Checking Correctness Of Experiments:**

N/A

**Review Assessment: Thoroughness In Paper Reading:**

I made a quick assessment of this paper.

---

> ### Author Response · Authors · 2019-11-13
> **Response to Reviewer 3**
>
> Thanks for the positive feedback and suggestions.
> We will further revise our paper according to your suggestions to make our contributions clearer.

---

### Official Review · AnonReviewer2 · 2019-10-23
**Official Blind Review #2**

**Rating:** 3

**Review:**

This paper studied the generalization performance of gradient descent for training over-parameterized two-layer neural networks on classification problems. The authors proved that under a neural tangent based separability assumption, as long as the neural network width is $\Omega(\epsilon^{-1})$, the number of training examples is $\tilde\Omega(\epsilon^{-4})$, within $O(\epsilon^{-2})$ iterations GD can achieve expected $\epsilon$-classification error.

Overall this paper is well written and easy to follow. The theoretical results on the neural network width and iteration complexity are interesting.

My major concern is that the comparison with Allen-Zhu et al and Cao & Gu seem somewhat unfair. First, Allen-Zhu et al and Cao & Gu both studied the generalization performance of GD for training multi-layer neural networks, which is fundamentally more difficult than two-layer networks. Second, they use ReLU activation functions, which brings in the nonsmoothness along the optimization trajectory. This would also make the condition on the neural network width become worse. Therefore, when claiming the advantage of the derived guarantees, the authors should clearly clarify such differences.

Another concern is that whether the derived theoretical results can be generalized to ReLU network?

When proving the generalization result, this paper takes advantage of margin-based generalization error bound. However, the generalization results in Cao & Gu are proved via applying standard empirical Rademacher complexity based generalization error bound. I would wonder which technique can give a tighter bound?

Can you provide some examples regarding which type of data can satisfy Assumption (A.4) with constant margin $\rho$?

 The authors would like to briefly discuss another data separation assumption adopted in the following papers (although this assumption is typically made for regression problem).

[1] Zeyuan Allen-Zhu, Yuanzhi Li, and Zhao Song. A convergence theory for deep learning via overparameterization. arXiv preprint arXiv:1811.03962, 2018b.
[2] Allen-Zhu, Z., Li, Y. and Song, Z. (2018c). On the convergence rate of training recurrent neural networks. arXiv preprint arXiv:1810.12065 .
[3] Difan Zou, Yuan Cao, Dongruo Zhou, and Quanquan Gu. Stochastic gradient descent optimizes over-parameterized deep relu networks. arXiv preprint arXiv:1811.08888, 2018.
[4] Samet Oymak and Mahdi Soltanolkotabi. Towards moderate overparameterization: global convergence guarantees for training shallow neural networks. arXiv preprint arXiv:1902.04674, 2019.
[5] Difan Zou and Quanquan Gu. An improved analysis of training over-parameterized deep neural networks. arXiv preprint arXiv:1906.04688, 2019.


**Experience Assessment:**

I have published one or two papers in this area.

**Review Assessment: Checking Correctness Of Derivations And Theory:**

I assessed the sensibility of the derivations and theory.

**Review Assessment: Checking Correctness Of Experiments:**

I assessed the sensibility of the experiments.

**Review Assessment: Thoroughness In Paper Reading:**

I read the paper at least twice and used my best judgement in assessing the paper.

---

> ### Author Response · Authors · 2019-11-13
> **Response to Reviewer 2**
>
> We thank the reviewer for the feedback. As you said, our problem setting is different from those in [Allen-Zhu et al. (2018a)] and [Cao & Gu (2019a)] and our theory is restricted to two-layer network with smooth activation functions. We think the contributions of their papers are nice and they are not included in our study. So, we have clearly stated this difference (depth and activation functions) in the revised version to clarify each position in this context. Thank you for the suggestion.
>
> A margin-based generalization bound is useful when the convergence is shown only for the empirical classification error. In general, to show the convergence for the logistic function is somewhat difficult compared to the empirical classification error due to lack of the strong convexity, but derived generalization bound on the expected classification error is comparable with the standard generalization bound (if ignoring the margin effect).
>
> There are a lot of examples that satisfy Assumption (A.4) because a tangent model in (A.4) includes a usual infinite-width two-layer network. Thus, Assumption (A.4) with a certain positive constant $\rho$ is satisfied as long as a data distribution is separable by an infinite-width two-layer network with mild weights $w(\theta)$. Note that this network can separate any region due to the universal approximation ability. We have emphasized this point in the revised version.
>
> As for the assumption on the data distribution. We note that a data separation assumption in [1-5] is essentially the same as the positivity of the NTK as shown in [5] (see Proposition 3.6 in arXiv ver. of [5]). On the other hand, our separability assumption is weaker than the positivity of NTK as stated in Proposition 1, that is, we do not need the positivity of NTK on the whole space. Thus, our theory does not require a separation of the training dataset like [1-5] made for the regression problem.
>
> Besides the above comments, as commented in another post, we have included an additional result in the revised version, which improves the sample complexity with an efficient network width by slightly refining the proof. In this result, we utilize the convergence of loss function. Please see that post for detail.

---

### Official Review · AnonReviewer1 · 2019-10-23
**Official Blind Review #1**

**Rating:** 3

**Review:**

This paper studies the training of over-parameterized two layer neural networks with smooth activation functions. In particular, this paper establishes convergence guarantee as well as generalization error bounds under an assumption that the data can be separated by a neural tangent model. The authors also show that the network width requirement in this paper is milder than the existing results for ReLU networks.

In terms of significance, I think this paper is slightly incremental. As is discussed in the paper, results on both convergence and generalization have already been established in earlier works even for deep networks. The major contribution of this paper is probably the weaker requirement of network width, as is shown in Table 1. However, all other results in Table 1 are for ReLU networks, and it has been discussed in Oymak & Soltanolkotabi, 2019 that the over-parameterization condition for smooth activation functions are naturally weaker than that for ReLU networks. Although Oymak & Soltanolkotabi, 2019 did not study generalization, based on their discussion, the result in this paper is not surprising. Moreover, the authors should probably add comparison to Cao & Gu, 2019b in Table 1.

Moreover, the results in this paper is restricted to two-layer networks with fixed second layer weights. This seems to be a much simpler setting than many existing results. The definition of neural tangent kernel in equation (5), as a result, seems to be over simplified, compared to the original definition given in Jacot et al., 2018. The improvement of requirement in network width, which is the major contribution of this paper, might not be very meaningful if it only works for shallow networks.


**Experience Assessment:**

I have published one or two papers in this area.

**Review Assessment: Checking Correctness Of Derivations And Theory:**

I carefully checked the derivations and theory.

**Review Assessment: Checking Correctness Of Experiments:**

N/A

**Review Assessment: Thoroughness In Paper Reading:**

I read the paper thoroughly.

---

> ### Author Response · Authors · 2019-11-13
> **Response to Reviewer 1**
>
> We thank the reviewer for the feedback. As you pointed out, [Oymak & Soltanolkotabi (2019)] also studies the over-parameterized network with the smooth activation function. However, their analysis is tailored to the squared loss function, so that direct comparison seems difficult. Generally speaking, the logistic loss is more challenging than the squared loss from the viewpoint of the optimization and generalization analyses because we cannot utilize the strong convexity (i.e.,  the linear convergence) and parameters will diverge. However, a much reasonable generalization bound can be obtained for the logistic loss compared to the case of the squared loss for the classification setting as shown in our study. This is because a separability assumption works effectively for the logistic loss while a more stronger assumption (i.e., the positivity of the NTK) is essentially required for the squared loss. Indeed, [Oymak & Soltanolkotabi (2019)] uses the positivity of NTK (see comment Reviewer 2). This is why their theory is not applicable to our setting and it has not been well studied that how small network width is sufficient for the classification problems with the logistic loss function. We would like to emphasize this point in the final version.
>
> We have added a comparison to [Cao & Gu (2019b)] in the revised version. In addition, we have included an additional result that shows an improved sample complexity of $O(\epsilon^{-2})$ with an efficient network width $O(\epsilon^{-3/2})$ by refining our proof. An additional proof technique is interesting but simple, so it might be useful for the community. Please see another post for detail.
>
> As you said, we focus on two-layer networks with a fixed second layer. However, this setting is essentially important to investigate the convergence behavior of the optimization method for the non-convex problems, and many studies (for instance [Du et al. (2019), Arora et al. (2019), Wu et al. (2019), Chizat & Bach (2018)], etc.) have been also considering the same setting. Therefore, we think two-layer networks are still an interesting research subject.

---

### Author Response · Authors · 2019-11-13
**Our paper has been revised.**

Dear reviewers,
We have updated the paper. The main changes are as follows.
1. Add a comparison to [Cao & Gu (2019b)].
2. State the difference with [Allen-Zhu et al. (2018a)] and [Cao & Gu (2019a)] clearly, that is, we clarify that these study cover deep ReLUs (which are challenging problem) and do not include each other because of the difference of the problem setting and network structure.
3. Explain that there are a lot of examples such that Assumption (A4) is satisfied.
4. Add a result that achieve a improved sample complexity $O(\epsilon^{-2})$ with an efficient network width $O(\epsilon^{-3/2})$.

As for a new additional result.
This result can be easily obtained based on our result as follows:

Step 1. Show the convergence of the loss function (Theorem 3 and Corollary 2) based on the convergence analysis of the functional gradient norm (which is a result in the previous version).
A proof for this statement is not difficult and a slight modification of well-known technique in the convex optimization literature (which is also utilized in [Allen-Zhu et al. (2018a)] and [Cao & Gu (2019b)])

Step 2. Provide a sharper bound (Proposition 2) on $\|\Theta^{(T)} - \Theta^{(0)}\|_2$ by utilizing the convergence of the loss function.
As a result, Rademacher complexity is much reduced and an improved sample complexity is achieved.

This additional proof is simple and easy to follow. However, this leads to a significant improvement of the sample complexity at the price of slight increase of network width: $O(\epsilon^{-1})$ -> $O(\epsilon^{-3/2})$.
In addition, a proof technique in Step 2 is quite interesting and might be useful for the community.

We would be grateful if reviewers would see the revised version.
We will also include other suggestions in the final version.

---

> ### Author Response · Authors · 2019-11-15
> **Our paper has been revised.**
>
> We have also added the following comments to the revised version:
> 5. Different proof techniques for the squared loss and the logistic loss functions.
> 6. Difference with another separation assumption in [1-5] (in review #2) made for regression problems.

---

### Decision · Program_Chairs · 2019-12-19

**Decision:**

Reject

**Comment:**

This article studies gradient optimization for classification problems with shallow networks with smooth activations, obtaining convergence and generalisation results under a separability assumption on the data. The results are obtained under much less stringent requirements on the width of the network than other related recent works. However, with results on convergence and generalisation having been established in other previous works, the reviewers found the contribution incremental. The responses clarified some of the distinctive challenges with the logistic loss compared with the squared loss that has been considered in other works, and provided examples for the separability assumption. Overall, the article makes important contributions in the case of classification problems. However, with many recent works addressing challenging problems in a similar direction, the bar has been set quite high. As pointed out by some of the reviewers, the contribution could gain substantially in relevance and make a more convincing case by addressing extensions to non smooth activations and deep models.